# Fatigue Analysis of a Jacket-Supported Offshore Wind Turbine at Block Island Wind Farm

**DOI:** 10.3390/s24103009

**Published:** 2024-05-09

**Authors:** Nasim Partovi-Mehr, John DeFrancisci, Mohsen Minaeijavid, Babak Moaveni, Daniel Kuchma, Christopher D. P. Baxter, Eric M. Hines, Aaron S. Bradshaw

**Affiliations:** 1Department of Civil and Environmental Engineering, Tufts University, Medford, MA 02155, USAjohn.defrancisci@tufts.edu (J.D.); mohsen.minaeijavid@tufts.edu (M.M.); dan.kuchma@tufts.edu (D.K.); eric.hines@tufts.edu (E.M.H.); 2Depts. Ocean/Civil and Environmental Engineering, University of Rhode Island, Kingston, RI 02881, USA; cbaxter@uri.edu (C.D.P.B.); abrads@uri.edu (A.S.B.)

**Keywords:** virtual sensing, modal expansion, fatigue, offshore wind turbine, structural health monitoring, damage prognosis

## Abstract

Offshore wind-turbine (OWT) support structures are subjected to cyclic dynamic loads with variations in loadings from wind and waves as well as the rotation of blades throughout their lifetime. The magnitude and extent of the cyclic loading can create a fatigue limit state controlling the design of support structures. In this paper, the remaining fatigue life of the support structure for a GE Haliade 6 MW fixed-bottom jacket offshore wind turbine within the Block Island Wind Farm (BIWF) is assessed. The fatigue damage to the tower and the jacket support structure using stress time histories at instrumented and non-instrumented locations are processed. Two validated finite-element models are utilized for assessing the stress cycles. The modal expansion method and a simplified approach using static calculations of the responses are employed to estimate the stress at the non-instrumented locations—known as virtual sensors. It is found that the hotspots at the base of the tower have longer service lives than the jacket. The fatigue damage to the jacket leg joints is less than 20% and 40% of its fatigue capacity during the 25-year design lifetime of the BIWF OWT, using the modal expansion method and the simplified static approach, respectively.

## 1. Introduction

In 2021, the United States of America (U.S.) set an ambitious target to install 30 GWs of OWTs by 2030. In addition to the 30 GW target, the U.S. Department of Energy (DOE) is funding research with the goal of installing 15 GWs of floating OWT by 2035. These targets are steps towards a long-term goal of 110 GWs of OWT by 2050 [1]. As of the summer of 2023, 932 MWs were under construction between the Vineyard Wind I and South Fork projects [2]. These two large-scale commercial projects represent a turning point for the US offshore wind industry, as it moves beyond small-scale demonstrations toward commercial-scale wind farms that will require advanced operations and maintenance processes. In their 2022 Offshore Wind Strategies Report, the DOE presented a list of factors that they expect to drive cost reductions for fixed OWTs, and over 40% of the decrease in the levelized cost of electricity was expected to come from operations and maintenance improvements [3]. Specifically, the DOE report mentions remote monitoring and improved decision-making tools to optimize the timing of maintenance actions as ways to improve turbine availability and reduce the number of person hours at sea.

Within an offshore wind farm, several critical components need to be maintained over the 25-year lifespan. In a 2019 failure mode effects and criticality analysis (FMECA), Scheu et al. 2023 identified 337 failures that could benefit from the application of monitoring systems [4]. About one-third of these failures were identified as a part of the tower and substructure of a wind farm. Based on the research from Scheu et al. 2023, it appears that all parts of the support structure, as defined by Det Norske Veritas (DNV), are potential candidates for remote monitoring systems. 

Structural health-monitoring (SHM) techniques aim to assess the health states of structures to prevent catastrophic conditions. These methods involve data interpretation and early diagnosis of the damaged structural elements, followed by predictive maintenance; an automated and online strategy for damage detection was developed [5,6] that used a continuously monitored system (e.g., an automated SHM system for an OWT structure). SHM can also provide information about the current state of the structural condition [7,8]. SHM techniques include vibration-based methods that are used for several purposes, such as system identification of structures, finite-element (FE) model updating, input loads and parameter estimations, quantification of uncertainties influencing fatigue loadings, and fatigue damage. The process of model updating or the integration of digital twinning is to update the digital twin based on the measured response of the structure, in this case, an OWT that is continuously monitored by sensors. The authors have experience in system identification and digital twinning of the 6 MW Block Island Wind Farm OWT [9,10,11,12,13] and an operational 6 MW monopile [14,15]. They also completed the Block Island Structural Monitoring Project for the U.S. Bureau of Safety and Environmental Enforcement [16].

While there has been significant interest in SHM and digital twinning for OWT, the industry has not coalesced around a single broadly applicable standard or guidance document. For example, Ramboll (a global multi-disciplinary engineering, design, and consultancy company and the global market leader in the detailed design of foundation structures for OWTs) recently developed a framework for the use of digital solutions and structural health monitoring for the integrity management of offshore structures [17]. Their framework provides a coupling between the measurements obtained from SHM and a digital twin. In addition, international institutes and standard organizations, such as the International Organization for Standardization (ISO), American Petroleum Institute (API), and Norsk Sokkels Konkurranseposisjon (NORSOK) provide standards in which requirements and recommendations relative to in-service inspection, condition monitoring, and maintenance of wind turbines are defined [18,19]. The Norwegian Geotechnical Institute also provided guidelines for structural health monitoring for OWT towers and foundations [20]. While international institutes and organizations provide general guidance for offshore structures, only one document provides guidance on OWT structures, specifically the German VDI 4551 “Structure Monitoring and Evaluation of Wind Turbines and Platforms” standard [21]. While the VDI document has some advice on monitoring, it is not commonly referenced in the industry. One potential reason for the lack of an accepted general guideline for designing SHM systems for OWTs is that, in most countries, they are not required. Although SHM is rarely required, it is a valuable tool for reducing the levelized cost of electricity through improved operation and maintenance.

Fatigue damage is caused by the initiation and propagation of cracks in a material due to cyclic loading. Fatigue damage is a major concern for OWT support structures due to repeated cycles from wind and wave loads. The load cases for OWT support structure design are established in “IEC 61400 Wind energy generation systems–Part 3-1: Design requirements for fixed offshore wind turbines” [22]. These load cases include operational fatigue loads (design load case (DLC) 1.2), special conditions (i.e., DLC 2.4, DLC 3.1, DLC 4.1), and parked conditions (i.e., DLC 6.4). The fatigue-limit state is often a major focus for designers and researchers when studying OWT support structures [23,24,25]. A common challenge for fatigue-damage analysis on OWT support structures is that the process of the installation and wiring of sensors and data collection can be costly and, in some cases, the installation of sensors can be impossible at fatigue hotspots underwater, where a fatigue crack can be expected to initiate. It is often not possible to instrument below the water line, where fatigue damage must also account for the risk of corrosion. Moghaddam et al. 2019 studied corrosion pitting effects on fatigue crack-propagation behavior of floating offshore wind-turbine (FWT) foundations using a finite-element model of a spar-type FWT [26]. Pitting corrosion is known to significantly reduce the fatigue life of a structure depending on the seawater exposure time [27].

Recent studies on the foundation fatigue of wind turbines have focused on improving the understanding of structural integrity and the long-term performance of turbine foundations. For example, Shi et al. 2015 studied the soil–structure interaction on the response of an OWT with a jacket foundation. They employed two models: one with a flexible foundation with the p-y model considering the pile-group effect and another with a fixed base. They concluded that soil–structure interaction should be considered in the design and load calculation of jacket-supported OWTs. They also suggested that the pile-group effect for the jacket foundation should be considered during fatigue analysis [28]. Pimenta et al. 2024 proposed a new methodology to estimate bending moments and fatigue life estimation from acceleration data at the tower top of FWTs. They validated their method using experimental data from one of the three full-scale FWTs located at Wind Float Atlantic with a total 25 MW farm capacity [29]. Ma et al. 2024 investigated the effect of changes in damping, stiffness, and permanent accumulated rotation of monopile foundations due to cyclic loadings. They found that those changes resulted in a 10% and 16% decrease for the medium and ultimate states, respectively [30]. Mehmanparast et al. 2024 re-evaluated fatigue design curves for OWT monopile foundations by analyzing thick as-welded test data. They found that the inverse slope of the S-N curve for weldments can be higher than m = −3 [31]. In addition to conventional structural analysis methods, neural networks can predict structural responses nowadays. The emerging trends can be found in [32,33].

The limitations associated with actual data collection have inspired two critical studies for SHM of OWT structures: sensor placement and modal expansion. Optimal sensor placement has a long history going back to SHM for structures subjected to earthquake or wind loads [34]. These methods have been applied in the context of OWTs and show that the optimal placement of sensors is affected by both the support structure’s unique characteristics and the SHM campaign’s goal: parameter estimation or strain estimation [35]. Even with an optimal placement of sensors, an asset manager may want to estimate the fatigue damage at a non-instrumented location. In this case, virtual sensing methods can be used to estimate the strain at such locations. Virtual sensing is the process of estimating the response of a system at locations that are difficult to measure through methods that use a combination of models and existing physical sensor data.

The modal expansion method has been used for virtual sensing on monopile support structures and applied in practice on a 3 MW offshore wind farm [36]. Marius et al. 2020 tested the modal expansion method against small-scale laboratory tests representing large oil and gas platforms subjected to offshore wave loading [37]. In addition to the modal expansion method, Ziegler et al. 2016 studied an extrapolation algorithm to monitor the lifetime of an OWT [38]. Recent work has utilized the modal expansion method on numerical simulations of the OC4 jacket foundation to analyze the structural fatigue [39]. Although researchers have used modal expansion methods for several numerical models and large-scale monopiles, there is a gap in the field to investigate structural fatigue in OWT jacket foundations using long-term operational measurements. This study investigates the fatigue analysis for a GE Haliade 6 MW jacket-supported OWT using a full year of measurement data and a validated digital twin of the OWT. 

The fatigue-limit state is usually dominant in the design of the OWT foundations, and it is caused by a high number of load cycles experienced by the OWT during its lifetime. While the fatigue-limit state loads are calculated using the standards and guidelines for an OWT foundation, and it is designed for such loads. One may wonder how much of those loads are experienced by a currently operating turbine or if a currently operating turbine is experiencing lower or higher loads during its lifetime. Additionally, there is a gap in the literature about the fatigue analysis of a large-scale OWT with a jacket support structure using the modal expansion method. To answer those questions and fill the gap in research, in this paper, a digital twin of the B2 OWT within the Block Island Wind Farm (BIWF) is studied to assess actual loads and stress experienced by its foundation.

The studied OWT has been operating since 2016 and is instrumented with accelerometers and strain gauges [40]. The fatigue analysis helps us to assess the damage experienced by the B2 OWT and estimate the remaining lifetime of it. The fatigue demand on the foundation has been studied using a full year of data. FE models of the B2 turbine have been built in SAP2000, OpenSees, and OpenFAST tools, and verified in the authors’ previous studies [9,12]. Strain at several jacket joints is estimated using virtual sensing based on the FE model of the turbine. First, using a simplified static approach, strain-gauge measurements at the tower base are used to calculate the equivalent thrust load at hub height, and a SAP2000 FE model of the turbine is used to provide an estimate of the jacket joints’ stress due to the estimated equivalent thrust load. Second, strain at the virtual sensors at the jacket hotspots is predicted and compared to the simplified static approach. Then, structural fatigue analysis is performed using the predicted strain to calculate the damage to the B2 turbine during 1-year monitoring from 1 November 2021 to 30 October 2022 and estimate the remaining lifetime of the turbine. Finally, the effects of several environmental and operational conditions of the turbine are investigated on the damage of different jacket components. The considered operational parameters are yaw angle, pitch angle of the blades, rotor speed, power, wind speed, and ambient temperature.

## 2. BIWF Jacket-Supported OWT and Datasets

The monitoring system on B2 OWT of BIWF consists of nine wired accelerometers, four wireless accelerometers, eight strain gauges (SGs), and one inclinometer. The monitoring system, including the sensors, cables, and DAQ was designed and provided by the Norwegian Geotechnical Institute and installed by the authors with the help of General Electric technicians. The tri-axial accelerometers A1-A6 were installed in April of 2021 at three levels over the height of the tower: A1 and A2 at the height of 76.9 m, A3 and A4 at the height of 52.4 m above the deck, and A5 and A6 at the height of 27.9 m above the deck platform. Each pair, e.g., A1 and A2, were installed at the opposite inner surface of the tower at a specific height. The SGs were installed in October of 2021, and they have been providing measurements since then. They include four axial SGs and four circumferential SGs, which were paired one to one and installed on the inner side of the tower at about 0.5 m above the tower-to-deck connection bolts. The layout of the accelerometers and SGs that are used in the virtual sensing method is shown in Figure 1. The axial strain gauges that measure the strain along the tower’s vertical axis are labeled as SG45, SG135, SG225, and SG315 and located at 45°, 135°, 225°, and 315° from the platform’s west, respectively. They are also labeled $\varepsilon_{1},\varepsilon_{2},\varepsilon_{3},\varepsilon_{4}$ in the equations. Two coordinate systems are introduced: the (u, v) and the (X, Y). SG45 and SG225 are located along the v-axis, and SG135 and 315 are located along the u-axis. The Y-axis and the v-axis point to the platform’s north, and the magnetic north, respectively, as shown in Figure 1. The sampling frequency of the monitoring system is 50 Hz, and the data are stored in a series of 10-min-long data sets. More detailed information about the instrumentation process and sensors can be found in Hines et al. 2023 [40].

Due to the slight offset of SG45 from magnetic north, principal axes are defined as Mu along SG135 and SG315, as shown in Figure 1. Positive Mv is then defined as 90 degrees counterclockwise from positive Mu, which situates Mv between magnetic north and SG45. The light blue numbers, beginning with the number 1 positioned slightly clockwise from platform north, represent bolt numbers. During installation, the locations of SGs were noted according to bolt number, as shown in Figure 2. 

### 2.1. Dataset Selection

The data measurements used for this study are continuous vibration data taken from November 2021 to October 2022. Continuous time history measurements are saved in 10-min windows. Therefore, each dataset consists of 10-min SG time histories with a sampling frequency of 50 Hz (time step of 0.02 s). For every day, 144 datasets are available, and 4032, 4320, or 4464 datasets are available for every month, with 28, 30, or 31 days, respectively. Over a whole year, this resulted in 52,560 datasets, with some datasets missing due to the monitoring system being down. The total number of missing datasets, as shown in Table 1, is 3441, which gives 49,119 total 10-min datasets for this study. 

Data from the supervisory control and data acquisition (SCADA) system are also available for the BIWF-B2 turbine. The data give relevant information about the turbine’s performance. The SCADA data are an average of the data over 10-min intervals. SCADA data in April 2022 from 9 April 2022 at 6:10 p.m. to 21 April 2021 at 8:10 a.m. are missing.

To match the SCADA time with the DAQ time, the clock had to be adjusted. The DAQ clock had a 17 min delay from the real time (SCADA time) in March 2023, while the delay was 5 min in June 2022. The clock offset for 12 months is as follows. The clock offset is set to be zero for the first 4 months, November 2021 to February 2022; 10 min for the middle 4 months, March to June 2022; and 20 min for the last 4 months, July to October 2022. 

### 2.2. Correction of Strain-Gauge Zeros

The installed strain gauges are type HBWF-35-125-6-99UP-SS, which are temperature-compensated, full-bridge sensors. SG measurements need to be corrected by the subtraction of a constant to show the absolute strain values. These constant correction values are different for each SG. The correction values also depend on the specific loading condition of the B2 turbine at the time of SG installation. In addition, the SGs may be subjected to tension or compression during the installation process. Considering all the uncertainties in determining the initial absolute strain values, four steps that are needed to determine the strain correction values for all SGs and calculate the actual strain and stress values are as follows. 

Step 1: Calculate tower-base moments:

The tower-base moments (Mu and Mv) for each dataset are calculated as shown in (1) and (2).
(1)$$M_{u}=\frac{(\varepsilon_{SG45}^{m}-\varepsilon_{SG225}^{m})EI}{2y}$$
(2)$$M_{v}=\frac{(\varepsilon_{SG135}^{m}-\varepsilon_{SG315}^{m})EI}{2y}$$
where $\varepsilon_{SG45}^{m},\varepsilon_{SG135}^{m},\varepsilon_{SG225}^{m},\varepsilon_{SG315}^{m}$ are the strain measurements from the strain gauges SG45, SG135, SG225, and SG315, respectively. Superscript *m* stands for measurement. *E* is the modulus of elasticity of steel, *E* = 200 GPa; *I* is the moment of inertia of the tube section of the tower base, *I* = 3.469 m^4^; and *y* is the distance of the strain-gauge location to the neutral axis, *y* = 2.952 m. *M_u_* and *M_v_* are the tower-base moments about the *u*-axis, and *v*-axis, respectively. 

The bending moments in two perpendicular directions in the coordinates (*u*, *v*), as shown in Figure 1, can then be used to determine the bending moments in the fore–aft (FA) and side–side (SS) directions. The nacelle orientation defines the FA direction and SS is perpendicular to it. Moments in the *FA* ($M_{FA}$) and *SS* ($M_{SS}$) directions are calculated as follows:(3)$$\begin{array}{l} M_{FA}=M_{u}\cos\theta -M_{v}\sin\theta \\ M_{SS}=M_{u}\sin\theta +M_{v}\cos\theta \end{array}$$
where θ is the yaw angle starting at magnetic north and is positive clockwise.

Step 2: Fit a circle to the moments of an idling turbine.

The moments of an idling turbine in the Mu and Mv coordinates correspond to the self-weight moments from the overhang of the center of mass of the RNA. For an idling turbine that does not produce power, the rotor spins very slowly, or it does not spin at all. For an idling/parked turbine, the axial force on the tower base is considered to be the self-weight of the RNA, and the self-weight moment of the RNA on any tower section can be calculated as the weight of the RNA times the overhang distance. There are no operational induced loads on a parked/idling turbine, and the wind loads are negligible compared to the static self-weight moments and loads. The direction of the self-weight moment changes with any change in the location of the RNA, which is determined by the yaw angle of the turbine. So, the moments should create a circle of about zero center for an idling turbine; however, the moments calculated from the SG data have a non-zero center. The center coordinates (*u_c_* and *v_c_*) are the initial moments calculated from SG measurements that should be subtracted from the measured moments to give us the self-weight moments, shown in (4) and (5).
(4)$$\frac{\left(\varepsilon_{SG45}^{0}-\varepsilon_{SG225}^{0}\right)}{2}=\frac{u_{c}y}{EI}$$
(5)$$\frac{\left(\varepsilon_{SG315}^{0}-\varepsilon_{SG135}^{0}\right)}{2}=\frac{v_{c}y}{EI}$$
where *u_c_* and *v_c_* are the center of the circle fitted to the idling moments in the (*u*, *v*) coordinates. $\frac{y}{EI}=4.255\times 10^{-6}\;(\mathrm{MN}\cdot {m)}^{-1}$ for the B2 turbine. 

Step 3: Compute the axial strains due to self-weight.

The weight of the RNA and tower causes axial strain and stress on the tower base. The strain can be calculated using the beam theory:(6)$$\frac{\left(\varepsilon_{SG45}^{m}-\varepsilon_{SG45}^{0}\right)+\left(\varepsilon_{SG225}^{m}-\varepsilon_{SG225}^{0}\right)}{2}=\varepsilon^{axial}=-\frac{W_{self-weight}}{EA}$$
(7)$$\frac{\left(\varepsilon_{SG315}^{m}-\varepsilon_{SG315}^{0}\right)+\left(\varepsilon_{SG135}^{m}-\varepsilon_{SG135}^{0}\right)}{2}=\varepsilon^{axial}=-\frac{W_{self-weight}}{EA}$$
where $\varepsilon_{SG45}^{0},\varepsilon_{SG135}^{0},\varepsilon_{SG225}^{0},\varepsilon_{SG315}^{0}$ are the strain measurements from the strain gauges SG45, SG135, SG225, and SG315, respectively. *A* is the tower-base section area, and *A* = 0.7849 m^2^. $W_{self-weight}$ is the weight of the RNA and tower, $W_{self-weight}=430+367.7=797.7\;\mathrm{ton}$. $\varepsilon^{axial}$ is the self-weight-induced axial strain, $\varepsilon^{axial}=4.99\times 10^{-5}=49.9\;\mu \varepsilon$. From (6) and (7), it can be written:(8)$$\frac{\left(\varepsilon_{SG45}^{0}+\varepsilon_{SG225}^{0}\right)}{2}=4.99\times 10^{-5}+\frac{\left(\varepsilon_{SG45}^{m}+\varepsilon_{SG225}^{m}\right)}{2}$$
(9)$$\frac{\left(\varepsilon_{SG135}^{0}+\varepsilon_{SG315}^{0}\right)}{2}=4.99\times 10^{-5}+\frac{\left(\varepsilon_{SG135}^{m}+\varepsilon_{SG315}^{m}\right)}{2}$$

Step 4: Solve for the strain’s correction values.

The SG-based correction values can be solved using (4), (5), (8), and (9). The value of key parameters can be calculated as follows.
(10)$$\varepsilon_{SG45}^{0}=4.255\times 10^{-6}\cdot u_{c}+4.99\times 10^{-5}+\frac{\left(\varepsilon_{SG45}^{m}+\varepsilon_{SG225}^{m}\right)}{2}$$
(11)$$\varepsilon_{SG135}^{0}=-4.255\times 10^{-6}\cdot v_{c}+4.99\times 10^{-5}+\frac{\left(\varepsilon_{SG135}^{m}+\varepsilon_{SG315}^{m}\right)}{2}$$
(12)$$\varepsilon_{SG225}^{0}=-4.255\times 10^{-6}\cdot u_{c}+4.99\times 10^{-5}+\frac{\left(\varepsilon_{SG45}^{m}+\varepsilon_{SG225}^{m}\right)}{2}$$
(13)$$\varepsilon_{SG315}^{0}=4.255\times 10^{-6}v_{c}+4.99\times 10^{-5}+\frac{\left(\varepsilon_{SG135}^{m}+\varepsilon_{SG315}^{m}\right)}{2}$$
where $\frac{\left(\varepsilon_{SG45}^{m}+\varepsilon_{SG225}^{m}\right)}{2}$ and $\frac{\left(\varepsilon_{SG135}^{m}+\varepsilon_{SG315}^{m}\right)}{2}$ are the mean of all measured strains for an idling turbine.

Once the initial strain values from the previous step are obtained, the real strain values can be obtained using (14). Then, the stress values can be calculated using (15).
(14)$$\varepsilon_{i}^{r}=\varepsilon_{i}^{m}-\varepsilon_{i}^{0}$$
(15)$$\sigma_{i}^{r}=E\cdot \varepsilon_{i}^{r}$$
where $\varepsilon_{i}^{r}$ and $\sigma_{i}^{r}$ are the real (corrected) strain and stress values for the strain gauge i: SG45, SG135, SG225, and SG315.

## 3. Virtual Sensing

Using virtual sensing methods, strain at any section of a structure can be predicted using measured accelerations and strains at sensor locations and the finite-element (FE) model of the structure. In this paper, the following two approaches are used for virtual sensing: (1) a simplified approach to estimate the thrust load statically and (2) a modal expansion method. In the first approach, an equivalent thrust load is estimated and then used as the input to the FE model for virtual sensing. This approach tends to overestimate the damage and underestimate the service lifetime. In the second approach, modal expansion is used to predict the dynamic part of the strain signal while the quasi-static part of the signal is estimated using mode shapes. 

### 3.1. Simplified Approach Assuming Static Loading

Jacket fatigue is evaluated based on the stresses obtained from a SAP2000 FE model of the support structure for Turbine B2. This model is verified with the experimental data in the authors’ previous studies [9,12]. As shown in (17), the thrust force is calculated from the bending moments at the instrumented section and applied to the FE model. The resulting stresses on any point of the jacket foundation are determined via a static analysis of the turbine structure. As part of the correction process, the resulting self-weight moment of the RNA at the instrumented location is calculated as 17.72 MN·m. with (16).
(16)$$M_{self-weight}=m_{RNA}\cdot g\cdot d$$
(17)$$Thrust=\frac{M_{FA}+M_{self-weight}}{h}$$
where $M_{self-weight}$ is the self-weight moment of the RNA at the tower base, $m_{RNA}$ is the mass of the RNA as 430 tons, *g* is the gravity of 9.81 m/s, and *d* is the RNA overhang distance of 4.2 m. Then, the self-weight moment can be calculated as $M_{self-weight}=17.72\;\mathrm{MN}\cdot m$. $M_{FA}$ is the moment at the tower base in the fore–aft (FA) direction, and *h* = 82.85 m is the vertical distance between the RNA center of mass and the strain gauges near the tower base. 

To obtain the thrust for each yaw angle, a static load can be applied to the FE model for each time step, which will result in an estimate of the stress at any jacket node. 

For this paper, a 1 MN load is applied at the RNA level at four different angles, 0°, 15°, 30°, and 45°, as shown in Figure 3, to estimate the stresses in the jacket members. The largest axial stress, S_11_ in the main axis of each element, is obtained in eight stations around the perimeter of the section, as shown in Figure 3c. The stress hotspot for fatigue analysis should be chosen at welds between the braces and the tubular joint. Station 3 is chosen for the leg joints and the maximum stress between eight stations for the braces. The result for each angle is shown in Table 2. The maximum S_11_ between all the angles is chosen as the stress for the leg–brace joints, as shown in Table 3. The stresses are then used to estimate the stress linearly by scaling the thrust force from 1 MN to any other thrust force over time.

As shown in Figure 4, five hotspots on the jacket were analyzed with a focus on the leg and brace joints. Three critical joints were then selected out of the five based on the stress values. The chosen joints are joint #1: leg joint and joint #2: brace joint in seawater with cathodic protection and joint #5: leg joint in the splash zone without cathodic protection. 

### 3.2. Modal Expansion

In the modal expansion approach, the dynamic and quasi-static strains at locations of interest are estimated separately, and the total strain is estimated as the summation of these two components [41].
(18)$$\varepsilon_{p}(t)=\varepsilon_{p}^{QS}(t)+\varepsilon_{p}^{D}(t)\;\forall t$$
where $\varepsilon_{p}(t)$ is the predicted strain at the location of interest, $\varepsilon_{p}^{D}(t)$ is the dynamic strain response, and $\varepsilon_{p}^{QS}(t)$ is the quasi-static strain response.

The *quasi-static strain* is the quasi-static part of the strains and is estimated as a ratio of the first strain mode shape at the location of interest to that of a reference strain measurement. For this part, the strain measurements are low-pass filtered (below 0.11 Hz) to represent only the quasi-static part of the response. The quasi-static strain response is calculated as (19).
(19)εpQS(t)=ΦεpQSΦεmQS εmQS(t)    ∀t
where εmQS(t) is the quasi-static response of the actual strain-gauge measurements, ΦεpQS∈ℝnp×1 is the numerical quasi-static strain at n_p_ virtual sensor locations, and ΦεmQS∈ℝ is the numerical quasi-static strain component at the measured DOF. In the u direction, (19) is applied so as to capture the quasi-static vibrations in the two perpendicular directions (u and v) using the superposition; (19) can be extended to (20).
(20)εpQS(t)=ru. εm,SG315QS(t)+rv. εm,SG225QS(t)    ∀t
(21)ru=ΦεpQSΦεm,SG315QS
(22)rv=ΦεpQSΦεm,SG225QS
where εm,SG315QS(t) and εm,SG225QS(t) are the quasi-static response of the measured strain gauges SG315 and SG225, and ru and rv are the strain mode shape ratios of the first mode in the predicted location in the jacket member to the strain in the tower section along with the u axis, and v axis, respectively. Φεm,SG315QS and Φεm,SG225QS are the strain mode shapes at the strain gauges SG315 (along the u-axis) and SG225 (v-axis) locations.

*The dynamic strain* response, $\varepsilon_{p}^{D}(t)$ in (18), can be estimated using the modal expansion method. The measured accelerations are numerically double integrated to estimate the displacements. Before integrating the acceleration data, a bandpass finite impulse response (FIR) filter is applied to the measurements within the frequency range of 0.11 to 3.00 Hz. The obtained displacement $d_{m}(t)$ is then filtered with a high-pass FIR filter with a frequency of 0.11 Hz to remove the low-frequency trend created by the integration process.

Using the displacements at the measured DOF and the displacement mode shapes (e.g., first, second, and torsion modes) of the turbine from the finite-element (FE) model of the OWT, the modal coordinates of the OWT are calculated. The structural dynamic response, such as acceleration, velocity, or displacement, to any load can be written as a linear combination of responses in each eigenmode of the structure, written as (23).
(23)dm(t)=∑i=1Nϕi,mqi(t)     ∀t
where dm(t)∈ℝnm×1 is the displacement vector containing n_m_ measured locations, subscript m corresponds to the measured DOFs, ϕi,m∈ℝnm×1 is the i^th^ mode shape components at the measured DOFs, qi(t) is the modal coordinate component for mode *i* at time instance *t*. The mode shapes are obtained from the FE model of the OWT. Rewriting (23) in the form of matrices, dm(t) is: (24)dm(t)=Φmq(t)    ∀t
where q(t)∈ℝN×1 is the modal coordinates that represent the participation of each mode in the dynamic displacement response, containing N modal coordinate components for each time instance t. This is also written as q(t)=[q1(t),q2(t),…,qN(t)]T, and Φm∈ℝnm×N is the mode shape matrix at the measured n_m_ DOFs for N modes, Φm=[ϕ1,m;ϕ2,m;…;ϕN,m]T.

To compute the modal coordinates q(t), the solution of (24) for **q**(*t*) is written as:(25)$$q(t)={\left(\Phi_{m}^{T}\Phi_{m}\right)}^{-1}\Phi_{m}^{T}d_{m}(t)=\Phi_{m}^{†}d_{m}(t)\;\forall t$$
where ${•}^{†}$ is the pseudo-inverse of a matrix •. Note that (25) is valid if the number of measured DOFs n_m_ is equal to or greater than the number of considered modes N. In this regard, the number of sensors installed on the structure restricts the number of modes to be considered in modal expansion.

Using the estimated modal coordinates in (25), displacements can be predicted similar to the measured displacements in (24) and calculated in (26).
(26)dp(t)=Φpq(t)    ∀t
where Φp∈ℝnp×N is the mode shape matrix at the n_p_ virtual sensor locations, and subscript p corresponds to the predicted DOFs. The prediction for displacements is provided by (26); however, the goal of this paper is to predict strains. So, to predict strains from predicted displacements, the strain-mode shapes should be calculated. The dynamic strain response can be calculated with the strain mode shapes and the modal coordinates using (27).
(27)εpD(t)=Φεpq(t)    ∀t
where εpD(t)∈ℝnp×1 is the predicted strain at n_p_ predicted DOFs, and Φεp∈ℝ1×N is the strain mode shape at the predicted location for N modes. Combining (25) and (27) results in (28), which helps to skip using (26).
(28)εpD(t)=ΦεpΦm†dm(t)    ∀t

The strain mode shape, Φεp, needs to be calculated differently for the tower and jacket elements, because the strain in the tower sections comes from axial, shear, and bending deformations, while in jacket members, the only strain source is the axial deformation. For a jacket member, strain is calculated from the axial deformation over the initial length of an element. Calculating the strain mode shape in any location at a tower section (29) is used.
(29)Φεp=TfΦdp 
(30)Φdp=[u1(i)u2(i)u3(i)u4(i)u5(i)u6(i)]
(31)Tf=[−1L−f0(12fLL3−6L2)−f0(6fLL2−4L)1Lf0(12(L−fR)L3−6L2)−f0(6(L−fR)L2−2L)]
where Φdp∈ℝ6×N is the displacement mode shape matrix containing two translational DOFs u1(i),u2(i) and one rotational DOF u3(i) at the start and three similar DOFs (u4(i),u5(i),u6(i)) at the end of the element *i*. **T**_f_ is the transformation vector from displacement mode shapes (obtained from the FE model) to strain mode shapes [42], *L* is the length of the element, *f*_0_ is the distance from the neutral axis of the section to the strain-gauge location. *f_L_* and *f_R_* are the longitudinal distance from the left and right ends of the strain gauge to the left and right ends of the element, respectively. 

The evaluation metrics used in this paper are the time-response assurance criterion (TRAC), which can be found in [43], and the relative root mean square error (RRMSE) in the time domain, as written in (32). They are used to quantify the quality of the predictions vs. the measured signal.
(32)RRMSE=1ns∑t(εpmeas(t)−εppred(t))21ns∑t(εpmeas(t))2
where εpmeas(t)∈ℝ is the measured strain-gauge signal at the predicted DOF at time instance t, εppred(t)∈ℝ is the predicted strain at the predicted DOF p, subscript p implies the predicted DOF, and n_s_ is the number of data samples. RRMSE indicates the relative error, so a smaller number indicates a better match.

## 4. Fatigue Analysis

This section discusses the rainflow counting of the stress cycles and the fatigue lifetime assessment. The first step in assessing fatigue damage is to count the stress cycles and determine the stress ranges from a window of stress-response data. According to the ASTM E1049-85 standard [44], several counting methods are related to the fatigue-damage assessment, and rainflow counting is used in this paper. Referring to IEC 61400-3-1, rainflow cycle counting is a conservative method of counting cycles that is often used for fatigue design and assessment. In addition to rainflow counting, the IEC standard also allows mean cycle crossing methods to be used; however, rainflow cycle counting appears to be a commonly used method in the wind industry [45]. Therefore, the rainflow cycle counting method was selected for this study.

Rainflow cycle counting is defined in the ASTM E1049-85 Standard Practices for Cycle Counting in Fatigue Analysis [44]. Several programming environments such as Matlab have a rainflow cycle counting function built into them or available in their open-source library. This function uses the same algorithm defined in ASTM E1049-85. The result of rainflow counting is a series of full- and half-cycle counts associated with a stress-cycle range and a mean stress. Before running the rainflow algorithm, the data are run through a combination of hysteresis and peak-and-valley filtering [44]. These filters are applied to remove the “noise” peaks and valleys that are below 0.5 MPa.

Hysteresis filtering was used based on recommendations from the online resources related to fatigue analysis found in both the Siemens and MATLAB documentations [46]. Hysteresis filtering works by removing reversals below a minimum threshold from the time series. For this analysis, the minimum threshold is set at 0.5 MPa. After utilizing hysteresis filtering to remove these very small cycles, peak–valley filtering is used to identify the local minima and maxima in the time series. Hysteresis filtering and peak–valley filtering are found in the ‘findTurningPts’ function from the MATLAB resource on fatigue analysis. Hysteresis and peak–valley filtering reduce the low-stress cycles to 0.5 MPa, and the peaks and valleys of the stress time history become clearer. However, hysteresis filtering is insufficient, as it does not have a frequency component to reduce noise, and the low-stress cycles (e.g., 1 MPa) remain. So, an FIR filter added to the raw measurement is needed to remove low-frequency noises. In this study, FIR, hysteresis, and peak–valley filters are used to assess the fatigue damage. 

Fatigue damage and remaining life are typically assessed according to S-N curves developed based on laboratory testing of small-scale steel specimens. “S” stands for stress, and “N” stands for the number of cycles to failure. Acknowledging the existence of multiple S-N curves and fatigue design documents that are potentially relevant to this work, DNVGL-RP-C203 was selected as the S-N framework of choice [47]. Most existing research on offshore wind turbines has utilized the DNVGL family of recommended practice documents. Figure 5 presents a simplified example of fatigue analysis using “in air” S-N curves for different weld classes. For each stress-cycle bin, the total number of observed cycles is found via a rainflow cycle count, and then, a ratio is formulated with the number of observed cycles and the number of cycles allowed at that point in the S-N curve. These ratios are then added for each stress-cycle bin to find the total accumulated fatigue damage. The accumulated damage of one refers to the allowable fatigue capacity assuming a design fatigue factor (DFF) equal to one. 

One key feature of typical fatigue analysis with S-N curves is Palmgren–Miner’s rule. Palmgren–Miner’s rule assumes that fatigue-damage accumulation is path independent, i.e., the sequence of loads does not matter. This simplifying assumption allows an engineer to add the ratios without accounting for their placement in the time domain. The equation featured in Figure 5 is the following: (33)$$D=\sum_{i=1}^{k}\frac{n_{i}}{N_{i}}=\frac{1}{\bar{a}}\times \sum_{i=1}^{k}n_{i}\times {\left(\Delta \sigma_{i}\right)}^{b}\leq \eta$$
where D is the total damage by fatigue from the time history, $n_{i}$ is the number of cycles observed in that stress bin from the rainflow counting, k  is the total number of stress bins, and $N_{i}$ is the failure point on the S-N curve for the associated stress bin. The value of $N_{i}$ can be further broken down into the components of the S-N curve. $\bar{a}$ is the y-intercept of the S-N curve. It is often given as log($\bar{a}$) since the linear regression done to make the curve is performed in log space. b is the negative inverse slope of the S-N curve in log space. It is either three or five depending on the curve and position on the curve. η is the maximum allowable damage. This is typically one. However, in cases with a DFF, this may be 0.5 or 0.33.

For this analysis, the S-N curves used come from the DNVGL-RP-C203 recommended-practice document. B1 in air for a base material is selected for the tower. C1 in air is selected for tower welds that are assumed to be ground. Tubular and W3 welds were selected for the jacket. In addition to selecting these two weld types, two different environmental conditions are analyzed: in air and cathodic protection. Table 4 provides an overview of all the constants that define the S-N curves seen in (34). Based on the number of cycles N, the slope and intercept of either ($b_{1}$, $\log{\bar{a}}_{1}$) or ($b_{2}$, $\log{\bar{a}}_{2}$) is used for the S-N curve.
(34)$$\log\left(N\right)=\log\left(\bar{a}\right)-b\times \log\left(\Delta \sigma \right)$$

The curves applied in this study to the BIWF jacket fatigue assessment are the tubular joints with different environmental conditions and the W3 curves for the partial penetration welds because of the details of the welding at the joints. The assumption of the partial penetration is based on the fact that the backup weld is not subject to any inspection. This conservative assumption is made based on DNVGL-RP-C203 table A-10 for tubular members. For base material, a condition that is rarely relevant for fatigue, a tubular or T S-N curve can be used. If there is a partial penetration weld for a tubular member, a W3 curve is recommended. 

The environmental conditions from Table 4 are described as follows.
In-air curves represent material that is not exposed to any corrosive conditions. This includes parts of the structure far above the water, such as the tower, as well as material in the splash zone that is protected by an intact coating;Cathodic protection curves are used for material that is in the submerged zone of the structure and is protected with a cathodic protection system. Although corrosion information is not included in this study, it is reasonable to assume that steel submerged in water was designed to be cathodically protected. According to DNV-RP-0416, it is mandatory that external surfaces of the submerged zone have cathodic protection [50].

Finally, when performing a fatigue design analysis, there is an important DFF applied as a safety factor to account for a variety of uncertainties, including loading amplitudes, the potential for defects/corrosion damage, and the loading sequence. The design document DNV-ST-0126 Support Structures for Offshore Wind Turbines suggests using DFF = 2 for the members in the atmospheric zone with accessibility to inspection and DFF = 3 for the members in seawater with no inspection and repair accessibility [51]. 

## 5. Results

This section first discusses the results of strain correction and experimental studies using the modal expansion method. This is followed by the results of the fatigue lifetime analysis and the environmental and operational effects on the damage to the BIWF-B2 OWT.

### 5.1. Strain Correction Results

In an experiment on 30 June 2022, the BIWF-B2 turbine was manually rotated from 0 to 360° to capture the self-weight moments to correct the SGs. The turbine was initially set to be located at the north offset, i.e., yaw = 130°. Then, it was rotated manually in 10° increments using SCADA. The blades were fully feathered to keep the rotor from spinning and minimize any operational conditions on the experiment results. Note that the yaw angle starts from magnetic north, as shown in Figure 1, and rotates clockwise. The moments calculated from the raw SG measurements, as shown in Figure 6a, are on a circle as expected; however, the circle’s center is not at (0,0). So, the strain gauges are corrected so that the moments’ center is moved to the origin, as shown in Figure 6b. The raw strain-gauge measurements are corrected using (10), (11), (12), (13), and the circle center (*u_c_* and *v_c_*) values. The circle also does not match itself around coordinates (3,18). The slight mismatch is because the moment values in the plots are the average moments over 5 min of data with a standard deviation. 

It is important to note that SGs may not be suitable for long-term virtual sensing, as a drift in strain measurements can occur. The SG measurements undergo a correction process every month to address this issue in this study. In addition to the circle in Figure 7, eleven more circles are fitted for different months during the 1-year monitoring data. The radius for each month’s circle is checked to stay constant, as it represents the self-weight moment of the RNA and is not changed significantly over time. The center of the circles is kept at zero. Therefore, strain measurements are adjusted by checking the circles’ zero centers and radii to account for possible drift over time.

### 5.2. Experimental Study

This section uses the modal expansion method for strain prediction in instrumented and non-instrumented locations of the B2-OWT. Initially, the strain estimation is presented at the same instrumented locations of the tower-base section to further verify the modal expansion method. Then, the strain prediction is discussed in non-instrumented locations at different jacket structure components.

The strain mode shape, $\Phi_{\varepsilon p}$ in (29), used to predict strain at any location of the OWT is shown in Figure 7. At the tower base, the first bending mode in the X and Y directions plays a role in strain magnitude, whereas, at the tower top, the second bending mode in the X and Y directions has the most prominent strain. At the jacket level, both the first and second modes contribute to the strain measurements.

Initially, a study is done for one 10-min data window on 1 September 2022 starting at 02:19 am to do a supervised study. The SCADA data for this data set has a mean wind speed of 6.5 m/s, power generation of 1.2 MW, blades pitch angle of −1.5°, yaw angle of 270°, and wind misalignment of 6.5°. The accelerometer data are used at three levels of the tower (A1-A6, X and Y accelerometers), totaling 12 accelerometers and two strain gauges, SG225 and SG315, to estimate the strain at SG45 and SG135 locations, as shown in Figure 1.

#### 5.2.1. Strain Estimation at the Strain-Gauge Locations

To compare the experimental results for the tower-base section with the strain-gauge measurements, the strain-gauge measurements are split into two strain signals, namely one containing a frequency lower than 0.11 Hz (quasi-static part of the strain signal) and the other one with frequencies between 0.11–3 Hz (dynamic part), as discussed in Section 3.2. The dynamic part is estimated using the modal expansion method. For instance, the dynamic part of the SG135 signal, as shown in Figure 8, can be estimated with a TRAC of 0.994, indicating a great match. Then, the quasi-static part of the signal can be estimated using the negative of the counterpart strain gauge SG315, as shown in Figure 8b. Finally, summing up the dynamic and quasi-static responses, the total strain prediction would be as shown in Figure 8c. The TRAC and RRMSE between the measured strain gauge at SG135 and the predicted strain are 0.999 and 0.05, respectively, indicating another great match. Similarly, the SG45 can be estimated using the twelve accelerometers at the tower level and the strain gauge measurements of SG225. The TRAC and RRMSE between the measured and predicted strain are 0.995 and 0.05, respectively. 

The modal contribution from each mode in the dynamic strain response of SG135 is shown in Figure 9. Modes 1 and 2 correspond to the first mode in *v* and *u*, respectively. Mode 3 is the torsional mode, and Modes 4 and 5 correspond to the second mode in *v* and *u*. The largest modal contribution comes from the first mode in the u direction because SG135 is located in the positive u direction, and there is a little contribution from the second mode in the u direction. This was expected by looking at the strain mode shape in Figure 7. The strain mode shape at the tower base has a larger strain in the first mode than in the second mode.

#### 5.2.2. Strain Prediction at the Jacket Leg/Brace

Several hotspots on the jacket elements are selected, as shown in Figure 4. To predict strains on the jacket leg or brace, the ratios $r_{u}$ and $r_{v}$ are used in (21) and (22), as shown in Table 5. Using (20) and (30), the strain can be predicted at the jacket hotspots. The strain time history prediction for one 10-min window of data on 1 September 2022 starting at 02:19 am is shown in Figure 10.

#### 5.2.3. Strain Prediction on Jacket Using the Simplified Static Approach

Figure 11a,b show the measured moments at the tower base at the u-v and FA-SS coordinates, respectively. The estimated thrust corresponding to these moments is shown in Figure 11c. The estimated stress on the selected jacket joints (#1, #2, and #5) for one 10-min dataset in September 2022 is shown in Figure 11d. The dataset is selected based on the turbine’s operational condition. It is one of the datasets that has approximately maximum thrust force. The estimation is based on the product of the stress values in Table 3. 

#### 5.2.4. Comparison between the Simplified Static Approach and Modal Expansion 

To compare the strain time history prediction between the two methods, one 10-min data window is picked to compare the predicted strains, as shown in Figure 12. The shape of the two signals on the jacket leg elements matches. The TRAC for the jacket legs is more than 0.98, so the shape of the two predicted signals is similar. For the braces, the RRMSE is relatively high, and the TRAC is less than 0.40; however, the fatigue demand is low on the braces, which is discussed in the fatigue analysis in Section 5.3.2.

The predicted stress of jacket leg #5, obtained from Figure 12 by multiplying the strain by the modulus of elasticity of steel (200 GPa), is shown in Figure 13. The stress obtained from the simplified static approach is larger than that from modal expansion. The highest stress range is about 30 MPa using the simplified static approach, whereas it is about 20 MPa using the modal expansion method. The higher stress ranges cause greater damage prediction for the jacket nodes. For example, the total damage indices in this specific dataset are 7 × 10^−7^ and 5 × 10^−6^, using modal expansion and the simplified static approach, respectively. The damage calculated from the modal expansion prediction is about 10-times less than the simplified static approach. The damage indices are discussed further in Section 5.3.2.

The stress time history signal is split into dynamic and quasi-static signals to investigate where the error in the simplified static approach comes from, as shown in Figure 14. Both dynamic and quasi-static predictions have errors compared to the modal expansion method. The overall error comes from the simplified static approach, which does not fully consider the dynamic response of the structure and, as a result, overestimates the stress. The simplified static approach also assumes that the thrust force is applied from a direction that creates the largest stress in the jacket leg. So, this assumption makes the prediction relatively large compared to the modal expansion method.

### 5.3. Damage and Service Lifetime Estimation

In this section, the results from the fatigue analysis for the tower and jacket are discussed.

#### 5.3.1. Fatigue Assessment of the BIWF-B2 Tower

The data are filtered using the hysteresis method to evaluate the tower fatigue, and rainflow counting is used to evaluate the stress ranges and cycles. For example, the hysteresis filtering for one 10-min dataset is shown in Figure 15. After hysteresis and peak–valley filtering of the original data, as shown in a solid blue line with spikes and multiple stress values at peaks and valleys, is simplified to only include the single local peaks and valleys greater than 0.5 MPa. Rainflow counting could be used on the unfiltered dataset, but doing so would result in counting millions of low-stress cycles (e.g., 0.1 MPa) that do not have any meaningful impact on fatigue. A 3D histogram of these results is shown in Figure 16. 

Figure 17 shows breakdowns of the damage observed in each month at the strain-gauge locations of the tower base. The SG135 and SG315 have the most damage among all four strain gauges. The cumulative sum of all months gives the overall damage during 1-year monitoring. By linearly extrapolating the damage to the upcoming years, the lifetime can be estimated at the SG locations. Table 6 summarizes the estimated lifetime for the SG locations at the tower base from the cumulative damage during the 1-year monitoring. 

The results in the table show the impact of the fatigue limit. According to DNV C203, a detailed fatigue analysis can be omitted if the largest local stress range is below the fatigue limit at 1 × 10^7^ cycles. For the in-air curves, the fatigue limit for the B-1 and C-1 curves are 106.97 MPa and 65.50 MPa, respectively. As a result, fatigue damage is negligible for a B1 curve because all the stress cycles are below 100 MPa. However, for a C-1 curve, most of the cycles are below 65.60 MPa, but there are a few cycles above this limit, as shown in Figure 18. This means a fatigue analysis should still be done for the C-1 curve, but the results indicate that fatigue is not driving design because the lifetimes are over 10,000 years at all the strain-gauge locations. 

A review of the results in the preceding tables yields the following observations.
There is a clear sensitivity to the plate detail used in the fatigue analysis. Between a B1 curve representing a theoretical base material and a C1 curve representing a high-quality, ground, circumferential weld, there is an order of magnitude difference between 10^−6^ and 10^−5^;For the estimated 25-year lifetime, no significant amount of fatigue damage is observed in the tower. When estimating the remaining life in Table 6, the lowest value is 52,000 years.

Based on the observed data, it is unlikely that there would be any major fatigue concerns for the tower-base material at the location of the strain gauges, suggesting that fatigue concerns be directed toward areas of stress concentration, such as the flange details or to the bolts connecting the tower to the transition piece, which are known to experience tension losses over their lifetime. 

#### 5.3.2. Fatigue Lifetime in Jacket

The stress data for jacket components go through the same hysteresis filtering and rainflow counting process used for tower fatigue analysis. Two methods, discussed in Section 3, are used to compute the damage at three hotspots on the jacket foundation, and the results are compared. The cumulative damage over 1-year of monitoring for the jacket joints using different curves is shown in Table 7. Comparing the results obtained from the tubular joints (first two rows) and the W3 curves (the last two rows of the table) shows that fatigue damage is larger using the W3 curve than the tubular joint curve. The cumulative damage estimate for a 25-year lifetime of the B2 turbine jacket is shown in Figure 19. The results show that all the hotspots are below the 1:1 failure line. The modal expansion results show that jacket leg #1 in seawater has larger damage than leg #5 in the splash zone because the leg joint deeper in the water experiences larger stress than the hotspot located above the water. It can be concluded that the jacket joints will survive for 25 years of the turbine’s lifetime with damage of less than 20% using the modal expansion method. 

Table 8 provides a summary of the fatigue assessments from this study compared to the design estimates reported in the BIWF sub-structure and foundation design documents. The S-N curves are used to calculate fatigue. Using the curves, it is assumed that the welds and joints are inspectable (e.g., the joints welds in the splash zone). However, there are uncertainties associated with it, including defects and corrosion damage. In this regard, a DFF of two or three is used based on the environment. The lifetime estimation for jacket leg #1 is 196 years with a DFF of three using the modal expansion method. The design fatigue life for joint #1 reported is 26 years according to the design report, which is much less than the monitoring results (196 years from modal expansion). The simplified static approach, which is used to assess the fatigue life of the jacket using 1-year monitoring (e.g., 65 years for joint #1) is conservative, so the estimated remaining lifetime is smaller compared to the modal expansion results. 

The damage for jacket leg #5 is calculated using a W3 curve in air, assuming the coating in the splash zone remains intact for the full 25-year lifespan. The lifetime estimation for jacket leg #5 is 386 years for coated joints, considering a DFF of two, which is greater than the fatigue life of 50 years reported in the BIWF sub-structure and foundation design documents (Keystone, 2015). Finally, the lifetime estimation for jacket brace #2 yields an extremely low fatigue demand. Altogether, all three joints have an estimated service lifetime greater than the specified 25-year design life for the B2 turbine.

#### 5.3.3. Sources of Fatigue Damage in the Jacket

The analysis in Section 5.3.1 indicated that fatigue is not likely to be a driver of the tower base design. Nevertheless, a relative comparison can still provide perspective on what events, stress cycles, and conditions drive most of the observed fatigue damage. Figure 20 shows breakdowns of the damage observed in each month at the jacket hotspots. April and May 2022 have the highest damage for leg #1 and are almost twice as high as the next highest month. Note that April, the month with the highest observed damage, has the highest missing data, as shown in Table 1. Leg #1 in seawater has larger damage than leg #5 in air. 

Figure 21 is a cumulative plot showing how the fatigue damage accumulates across the month of April 2022, and it shows that there are sections with very steep slopes in April. This would seem to indicate that the damage accumulation in April comes from a few discrete events occurring rather than a high amount of damage occurring over typical operational conditions for the turbine. It can also be concluded that joint #5—leg in the splash zone—has more damage than joint #3—leg in the seawater. The brace does not have significant fatigue damage compared to the leg joints.

After observing that damage does seem to jump during events with some relation to time instead of keeping a constant slope, the next step was to analyze how the damage was associated with the stress bins. Figure 22 plots the damage for each 0.5 MPa stress bin throughout the year and April. Considering the full-year of data, there is a range from 10–30 MPa, where most damage occurs. The damage indices have a trend and less variability up to 20 MPa and then higher variability above 30 MPa. To get a better sense of when these large stress cycles are occurring, the stress cycles for April are shown. The stress cycles in April are within the range of 20–70 MPa. The higher range of stress-cycle counts, which have less damage in the full-year statistics, are discrete events happening to the turbine.

#### 5.3.4. Time Series Investigation

Based on the analysis of the months and the stress cycles, there appear to be at least two categories of operation, namely (1) stationary signals and (2) non-stationary signals caused by startup/shutdown. The stationary signals are observed to contribute to less damage than the startup/shutdown events. Moreover, some stress cycles provide higher damage, as shown in Figure 22. Comparing the plots in Figure 22 shows that there are discrete events happening in April 2022, when looking at the relative damage from high-stress cycles between 20 and 70 MPa. 

Additional scrutiny was given to the full-year data. There are interesting patterns in the stress time histories for the 10-min intervals with the highest damage indices. Figure 23 shows representative examples of non-stationary signals that have high damage indices events throughout the full year of monitoring, from November 2021, April, June, and September 2022. The November data show a yaw angle change. In September and June, clear “events” occurred and caused a large half cycle during the stress time histories. Based on the 10-min average SCADA data, this “event” appears when the blades’ pitch angle changes from −1.5° to fully feathered blades with a pitch angle of 90°. It is not possible to identify this event’s precise moment in the SCADA data, since the SCADA data is only in 10-min averages. However, the response of the structure looks like a classic free vibration with damping. The FA moment also changes greatly during the event, from +30 MN m to about −20 MN m. 

On 1 April, three cycles of about 30 MPa in SG45 and SG225 hold most of the damage. The SCADA data shows that the blade pitch was 23.8°, an average over the 10-min window of data, whereas the blades’ pitch angles were 11.9° and 35.7° before and after this event, respectively. The yaw angle is constant, but rotor speed and power generation were reduced in this event. This indicates that the pitch angle also plays a key role in the damage. The change in the pitch angle of the blades causes the operating turbine to change into an idling turbine (shutdown event), creating the high-stress cycles that happened on 12 September and 13 June 2022, as shown in Figure 23.

### 5.4. Environmental/Operational Effects on the Damage Index

In this section, the effects of the environmental/operational conditions, such as wind speed, rotor speed, power, ambient temperature, yaw angle, and pitch angle of the blades, on the damage index and cumulative damage of the jacket at joint #5 are investigated.

As shown in Figure 24, high damage indices (>10^−6^) occur when the 10-min average rpm is between 5–11.5 with different power generation levels, and the pitch angles are 0–40°. The histogram of pitch angles shows that the most frequent pitch angle is −1.5°, which is the pitch angle for the operating turbine at the rated rotor speed and rated wind speed. This pitch angle has the highest damage index. The less frequent pitch angle of 0–40° also brings high damage indices. These pitch angles are related to Region 3 of the power curve, with wind speeds between the rated (11.5 m/s) and the cut-out (25 m/s) wind speeds [9]. 

Figure 24f and Figure 25 show that the wind speeds correlated with the high damage indices in leg #1 and leg #5 are (1) wind speeds around 5 m/s when the turbine starts to operate (start-up condition) and (2) wind speeds around 10–11 m/s when the turbine produces the maximum power and the maximum thrust load. As shown in Figure 26, the high winds came mostly from SW and NW throughout the 1-year monitoring period, which is consistent with the damages that mostly occurred at yaw angles of approximately 230° and 320°, as shown in Figure 24e.

The highest damage index in Figure 24 is 1.1 × 10^−5^, and it is associated with a constant wind speed of about 11 m/s (rated wind speed) and a full capacity power generation of 6 MW, as shown in Table 9. The incident is the change in the pitch angle from 1° to 87° by looking at the nearest neighbors of this specific dataset. 

Another critical variable that impacts the damage index is the pitch angle of the blades. As shown in Figure 24c, the high damage values are also associated with the change in the pitch angle of the blades from 0 to 40° while the turbine operates. One theory for why high damage indices are associated with a shift in blade pitch is that, when the blades suddenly feather and reduce the thrust load, the tower moves in the fore–aft direction because of the eccentric position of the nacelle mass. Damage and change in the pitch angle for 1 year of monitoring data are shown in Figure 27. The zoomed-in plot for April is shown in Figure 28. The tower and nacelle oscillate freely without any thrust load from the feathered blades. Table 9 shows a 10-min dataset from 13 June 2022; while the turbine was operating at its rated rotor speed and generating full capacity, the blades were pitched and fully feathered. This could be considered an emergency shutdown or a maintenance shutdown, creating relatively high damage throughout the year. As shown in Figure 23, a half-stress cycle of about 95 MPa happened in an event on 13 June 2022. We suggest that, for maintenance purposes, operators do not shut down the turbine while it is operating at full capacity at a rated wind speed. The half-stress cycles throughout such events can create relatively large damage to the OWT foundation, whereas natural events (e.g., high winds) do not appear to damage the turbine support structure as much.

Although many factors affect the damage index (e.g., wind speed and rotor speed), the change in the pitch angle appears to play an important role. The damage index and the change in pitch-angle peaks occur at approximately the same time. The cumulative damage at legs #1 and #5 vs. the pitch angle during the 1-year monitoring is shown in Figure 29. There are some jumps in the cumulative plot, e.g., in May and early June 2022 that are related to the change in the pitch angle of the blades.

## 6. Conclusions

In this paper, fatigue analysis of the BIWF-B2 OWT substructure is investigated. The fatigue lifetime at the tower base and several hotspots at the jacket foundation is calculated for one year of monitoring data from 1 November 2021 to 30 October 2022. The strain gauges are corrected during an idling setting campaign on 30 June 2022 and used in the fatigue analysis. The fatigue lifetime at virtual sensors is calculated using the modal expansion method and a simplified static approach. The stresses predicted from the simplified static approach are larger, in general, than the modal expansion stress prediction. The error comes from the simplified assumptions made for the simplified static approach. The simplified static approach assumes that the thrust force is applied from a direction that creates the largest stress in the jacket elements. So, this assumption makes the prediction relatively large compared to the modal expansion method, which considers the directionality of the wind and the yaw angle. Consequently, damage in the jacket elements using the simplified static approach is higher than the modal expansion results. 

The results show that, for the estimated 25-year lifetime, no significant fatigue damage in the tower is observed. The lowest value is 52,000 years, which is negligible. The fatigue lifetime estimation for a jacket leg in seawater with cathodic protection is 196 years, considering a DFF of three and using the modal expansion method, while the fatigue design life is 26 years. The static method is used to assess the fatigue life of the jacket using 1-year monitoring (e.g., 65 years for joint #1) and is conservative, so the estimated remaining lifetime is less than the modal expansion results. The fatigue lifetime estimation for a jacket leg in the splash zone with coated joints is 386 years, considering a DFF of two, greater than the fatigue design life of 50 years. Note that DFF is used to account for uncertainties in fatigue-damage calculation, including defects or corrosion damage. The fatigue lifetime estimation for the jacket brace in seawater with cathodic protection yields an extremely low fatigue demand. Altogether, all jacket hotspots have an estimated service lifetime greater than the specified 25-year fatigue design life for the B2 turbine.

The high winds came mostly from SW to NW throughout the 1-year monitoring period. The wind speeds that are correlated with the high damage indices are wind speeds around 5 m/s when the turbine starts to operate (start-up condition) and wind speeds around 10–11 m/s when the turbine produces the maximum power and the maximum thrust load. The pitch angle of the blades has a great effect on the damage. The half-cycle stresses, which are followed by pitching blades from 0° to 90° in an operating turbine, cause relatively large damage to the turbine. The frequent pitch angle of 0–40° also brings the highest damage indices. The half-cycle stresses throughout such events can create relatively large damage to the OWT foundation.

As damage to the jacket leg is found to be larger than the one in the splash zone and maintenance is inaccessible underwater, particularly fatigue analysis and capacity should be considered when designing the foundation elements underwater. The hotspots at the splash zone are assumed to be coated for the remaining lifetime of the turbine, and they should be maintained coated. Otherwise, the damage would be greater than the estimated value in this paper. Moreover, to have fewer high-stress cycle ranges in the foundation hotspots, the number of shutdowns and startups should be minimized during the maintenance of offshore wind turbines. 

*Limitations and future work:* in assessing the fatigue damage, the effect of the thickness of a plate and the stress concentration factor (SCF) are not considered when using the S-N curves. The mean stress value is also dismissed from the fatigue-damage calculation. According to DNVGL-RP-C203, if part of the stress cycle is in compression, the stress ranges may be reduced by up to 20% before entering the S-N curve. So, considering the mean stress, thickness effect, and SCF is recommended for future work. Moreover, fatigue in the welded plates of the tower base is analyzed in this paper, but another important study is the fatigue damage of the bolts at the flange connections of tower segments and the connection between the tower and transition piece. For future work, a fatigue analysis of the bolts at the tower is suggested. Furthermore, this work has linearly extrapolated the 1-year fatigue damage to the 25-year lifetime of the OWT. For having an accurate assessment of the fatigue damage, a sophisticated model, such as a neural network, can be trained in future work to predict the fatigue damage using the forecast from climate models for the remaining design life for the BIWF site.

## Figures and Tables

**Figure 1 sensors-24-03009-f001:**
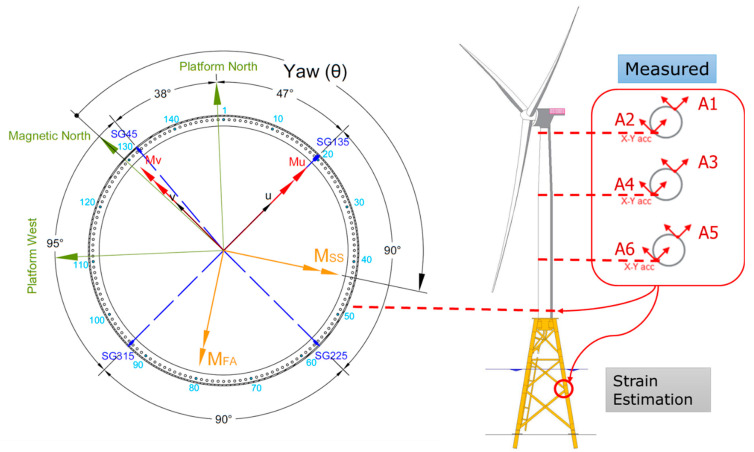
Instrumentation of the BIWF-B2 turbine, including the accelerometers on the tower and strain gauges at the tower base that are used in the modal expansion method. In the plan view of the platform, yaw is measured from the magnetic north.

**Figure 2 sensors-24-03009-f002:**
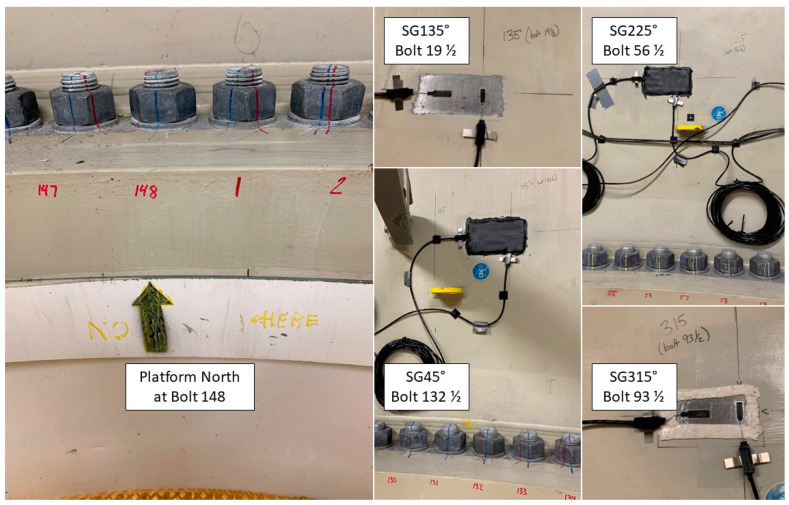
Turbine B2 strain-gauge locations according to bolt number.

**Figure 3 sensors-24-03009-f003:**
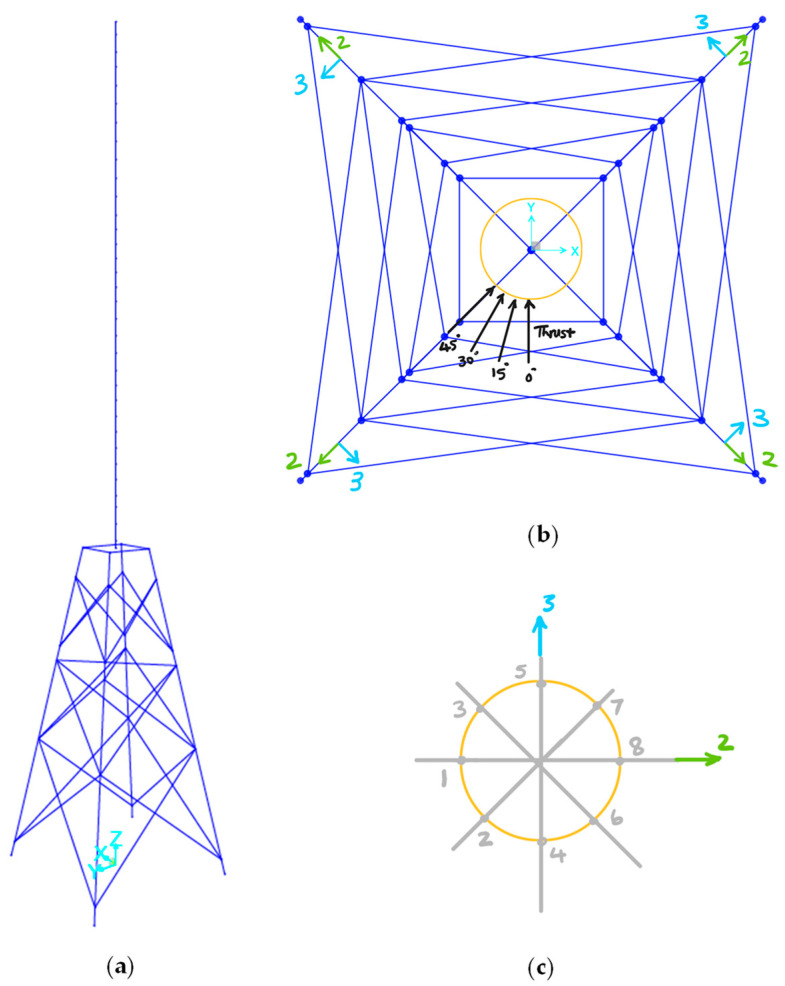
(**a**) 3D view and (**b**) top view of the BIWF-B2 turbine in SAP2000 tool (not to scale), and (**c**) one beam element section with 8 stations (#1 to #8). The local axes 2 and 3 of the jacket leg section in (**c**) are also shown in the global coordinates in (**b**).

**Figure 4 sensors-24-03009-f004:**
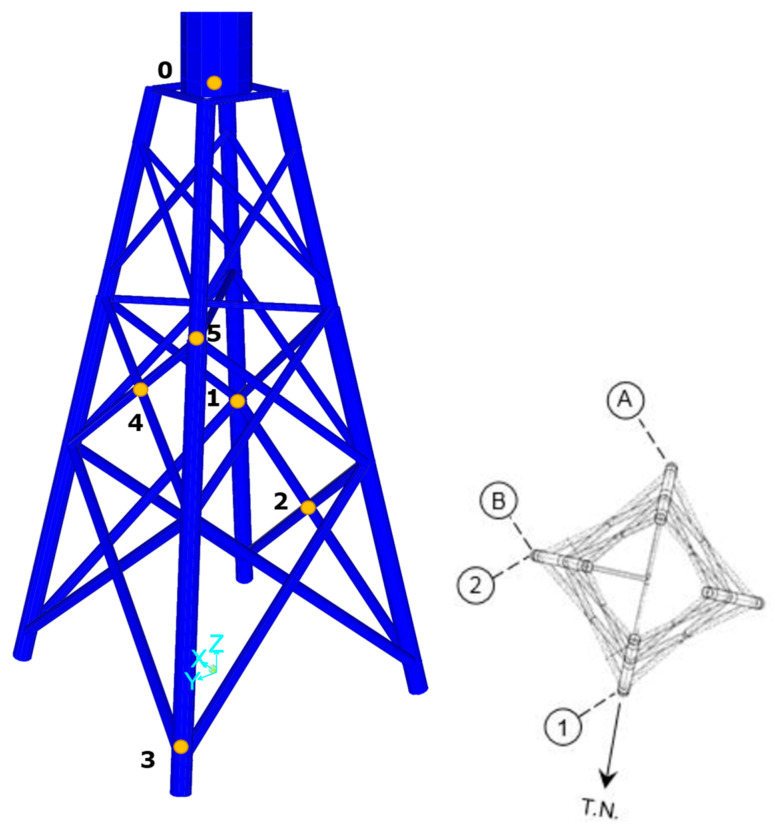
Hotspots on the jacket of the BIWF to predict strain. Node 0 is the measured strain at the tower base. Nodes 1 and 3 are the jacket leg nodes, and nodes 2 and 4 are the brace nodes in seawater with cathodic protection. Node 5 is the jacket leg node in the splash zone. Axis 1, 2, A, and B are the horizontal axis of the jacket leg at the mudline that were used in jacket drawing.

**Figure 5 sensors-24-03009-f005:**
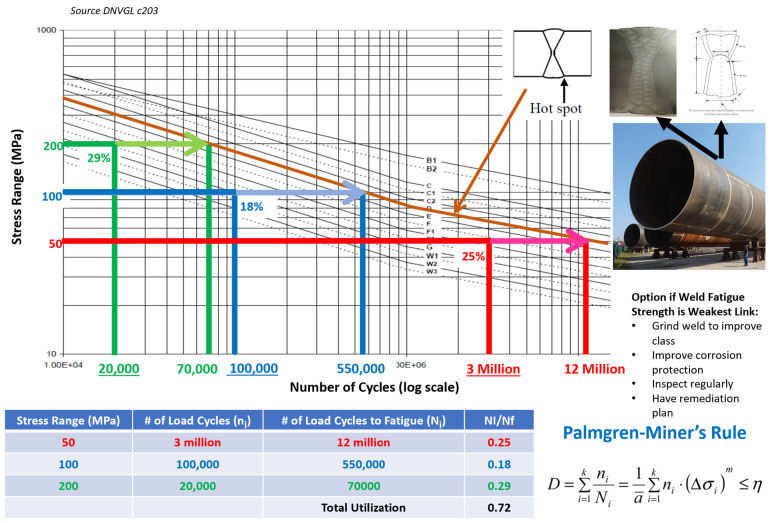
Example of S-N curve and simplified fatigue analysis, image sources [48,49].

**Figure 6 sensors-24-03009-f006:**
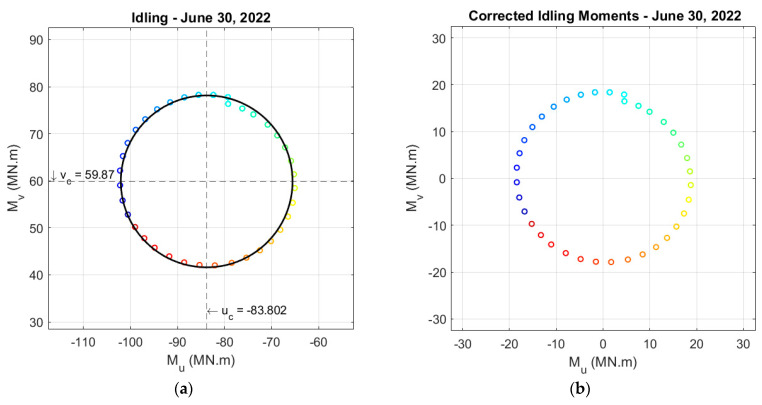
(**a**) Uncorrected moments and (**b**) corrected moments of the idling B2 turbine during the experiment on 30 June 2022. The color bar shows the change in yaw angle with 10° increments. The rainbow colors show yaw angles between 0–360° with the light blue indicating the zero-yaw angle.

**Figure 7 sensors-24-03009-f007:**
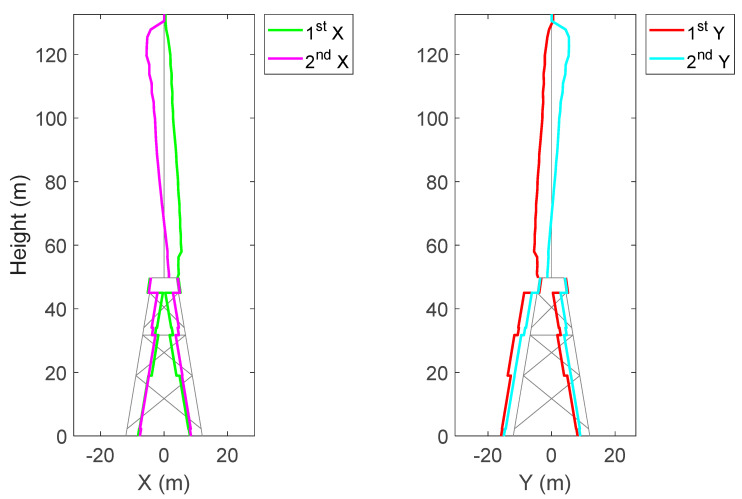
Strain mode shapes of the BIWF-B2 OWT for first and second bending modes in the X and Y directions.

**Figure 8 sensors-24-03009-f008:**
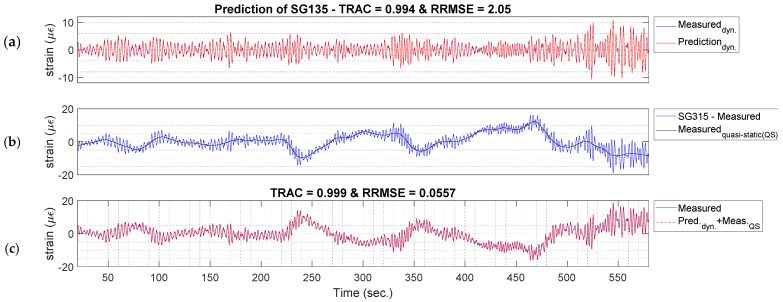
(**a**) Dynamic prediction of SG135, (**b**) measured quasi-static of SG315, and (**c**) Prediction of SG135 (dynamic prediction of SG135 plus measured quasi-static of SG135) at the tower base using the modal expansion method for a 10-min window of data on 1 September 2022 starting at 02:19 a.m.

**Figure 9 sensors-24-03009-f009:**
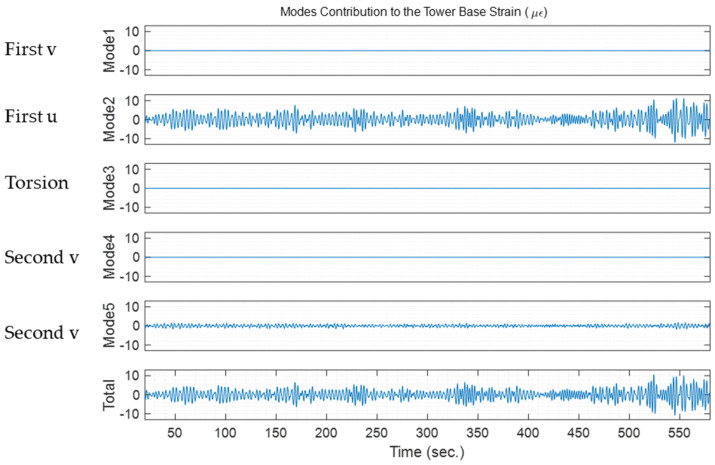
Modal contribution to the tower-base dynamic strain estimation of SG135 for a 10-min data window on 1 September 2022 starting at 02:19 a.m. Each signal for each mode is the $\Phi_{\varepsilon p}q_{i}(t)$ for *i* = 1, …, 5 modes. Total strain is the summation of the strain contribution from 5 modes.

**Figure 10 sensors-24-03009-f010:**
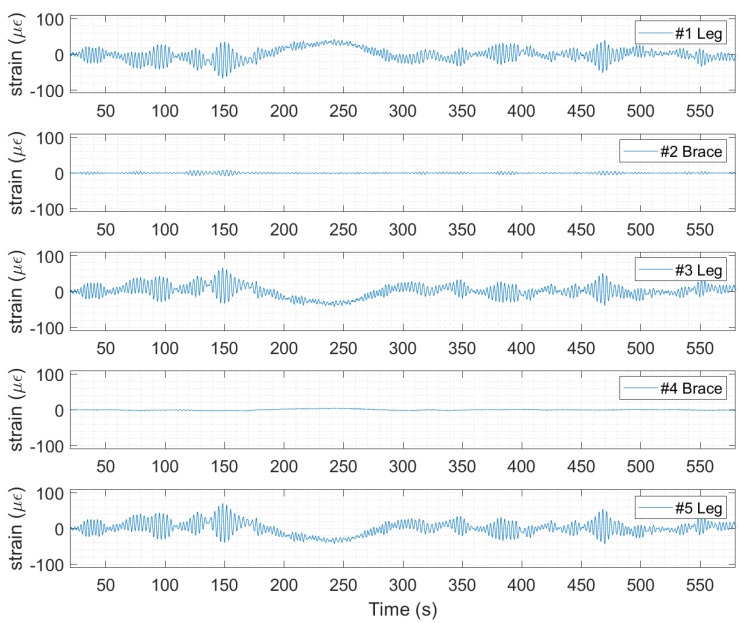
Strain prediction at jacket hotspots using the modal expansion method for a 10-min window of data on 1 September 2022 starting at 02:19 a.m.

**Figure 11 sensors-24-03009-f011:**
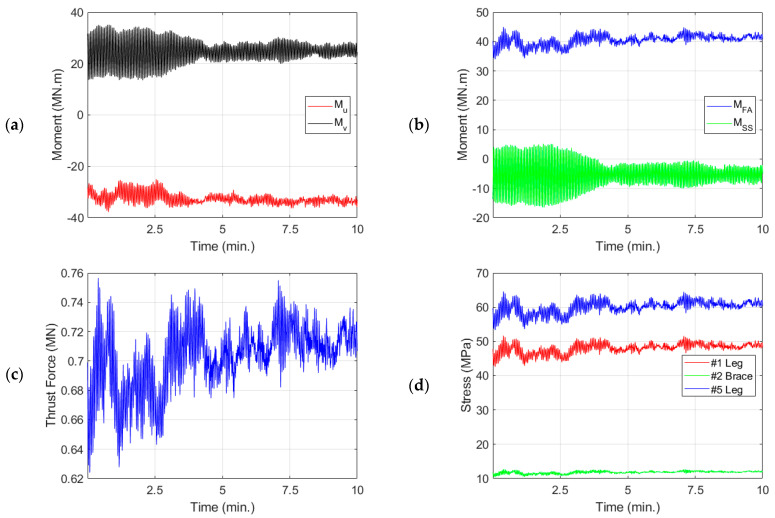
(**a**) Moments at (u, v) coordinates, (**b**) moments at FA and SS directions, (**c**) thrust force, and (**d**) jacket joints stresses for a 10-min interval of SG data starting at 26 September 2022 starting at 16:53 pm.

**Figure 12 sensors-24-03009-f012:**
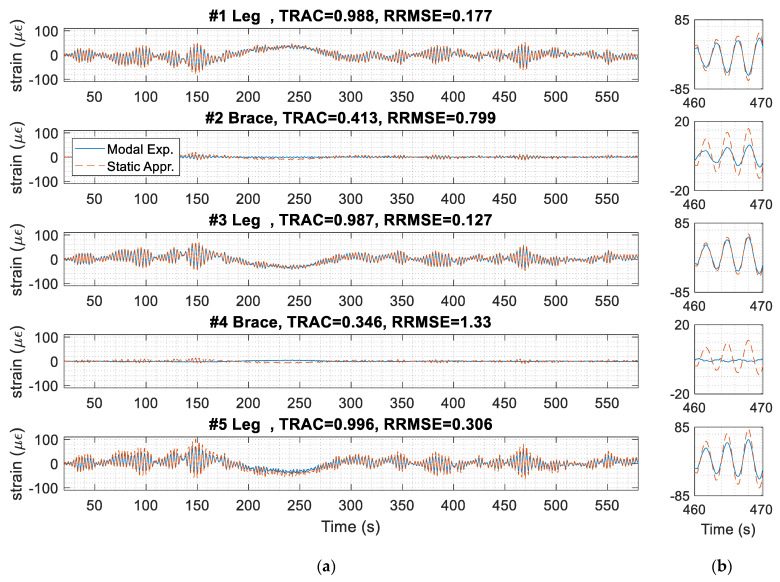
(**a**) Strain prediction at the jacket hotspots using the simplified static approach and the modal expansion method for a 10-min data window starting at 13:10 on 8 September 2022 with a yaw angle of 50°. (**b**) Zoomed-in signals between 460 and 470 s.

**Figure 13 sensors-24-03009-f013:**
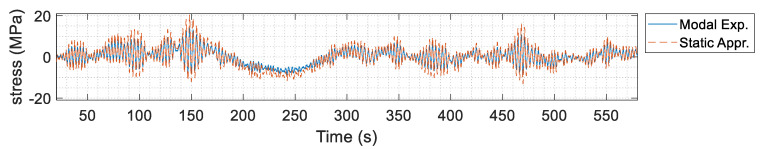
Stress prediction at jacket leg #5 using a simplified static approach and modal expansion method for a 10-min window of data starting on 8 September 2022 starting at 13:10 with a yaw angle of 50°.

**Figure 14 sensors-24-03009-f014:**
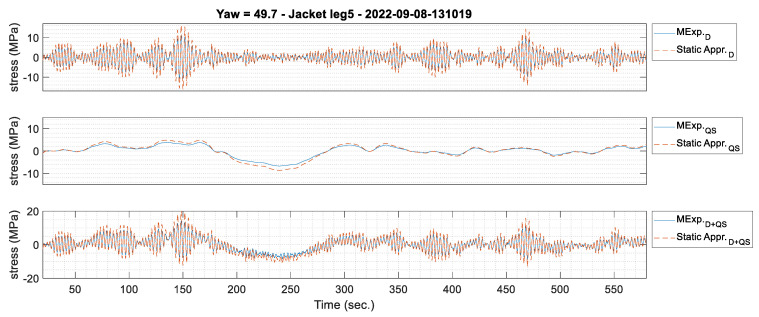
Stress prediction at jacket leg #5 using a simplified static approach and modal expansion methods for a 10-min data window starting at 13:10 on 8 September 2022 with a yaw angle of 50°; separated dynamic and quasi-static responses of the stress signal.

**Figure 15 sensors-24-03009-f015:**
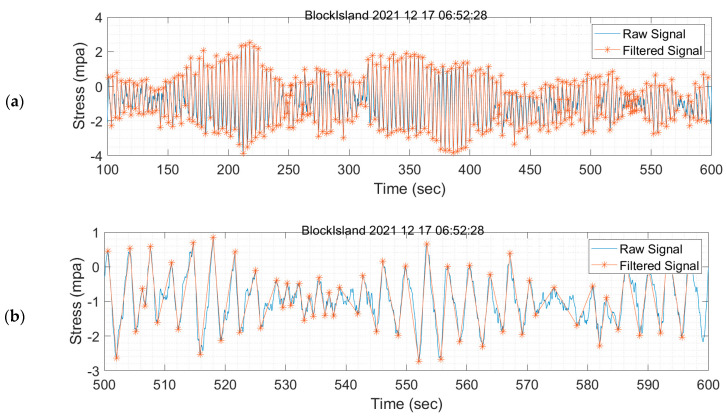
(**a**) Hysteresis and peak–valley filtering for one 10-min dataset on 17 December 2021 starting at 06:52:28 a.m. and (**b**) zoomed in between 500 and 600 s.

**Figure 16 sensors-24-03009-f016:**
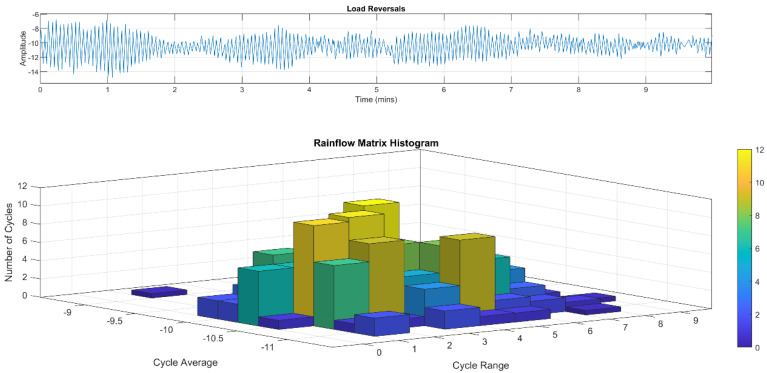
Rainflow counting for a 10-min interval on 17 December 2021 starting at 06:52 a.m.

**Figure 17 sensors-24-03009-f017:**
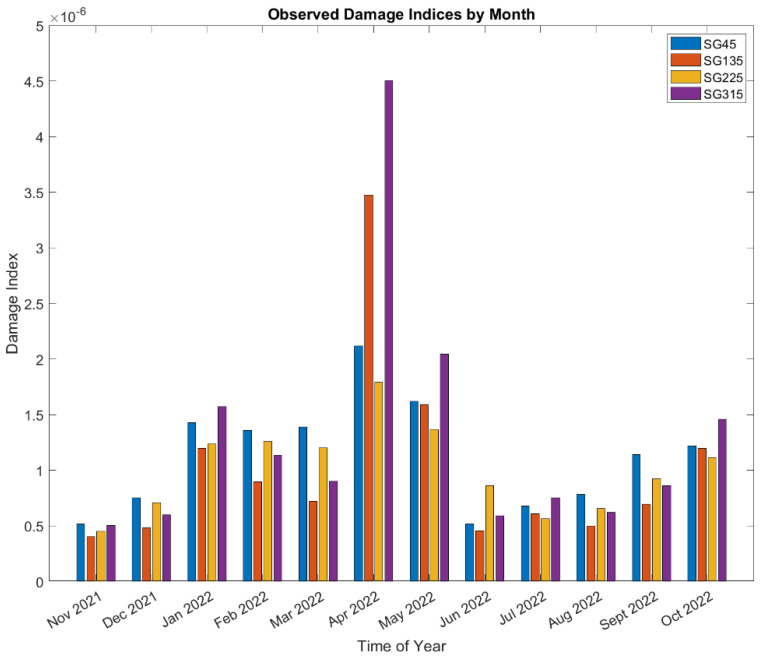
Damage indices breakdown by month at each strain-gauge location during the 1-year monitoring.

**Figure 18 sensors-24-03009-f018:**
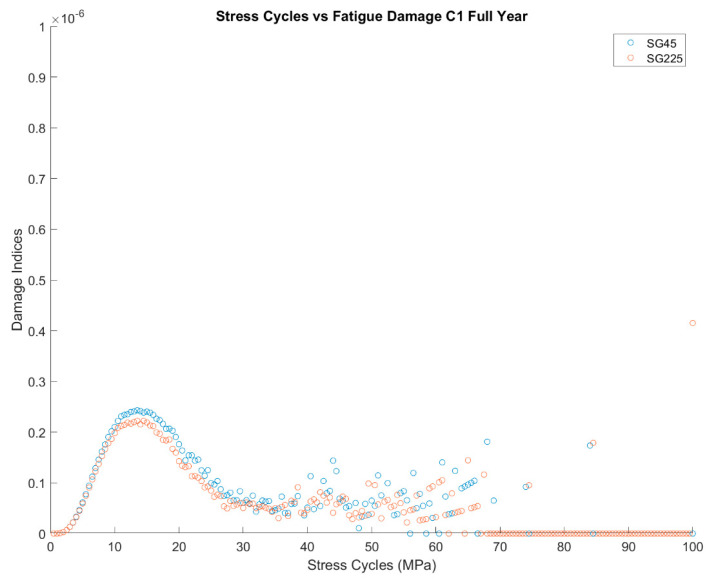
Damage by stress at the tower base for the entire year range.

**Figure 19 sensors-24-03009-f019:**
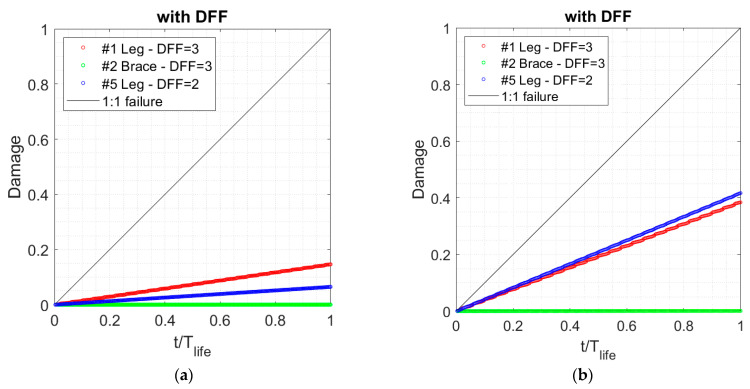
Damage accumulation over the lifetime of 25 years vs. time ratio to the lifetime of 25 years by extrapolating 1 y monitoring results and using (**a**) the modal expansion method, and (**b**) a simplified static approach. A DFF of 3 for the jacket leg in seawater and 2 for the leg in the splash zone are considered.

**Figure 20 sensors-24-03009-f020:**
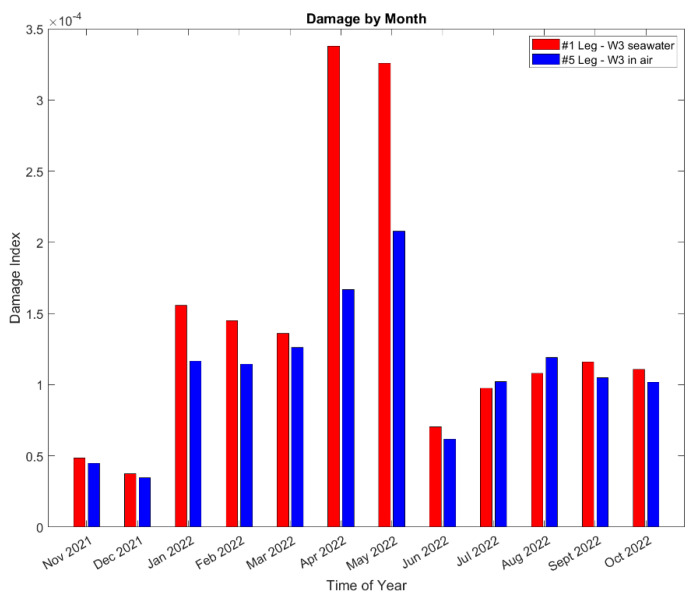
Damage at two jacket leg joints breakdown by month during the 1-year monitoring.

**Figure 21 sensors-24-03009-f021:**
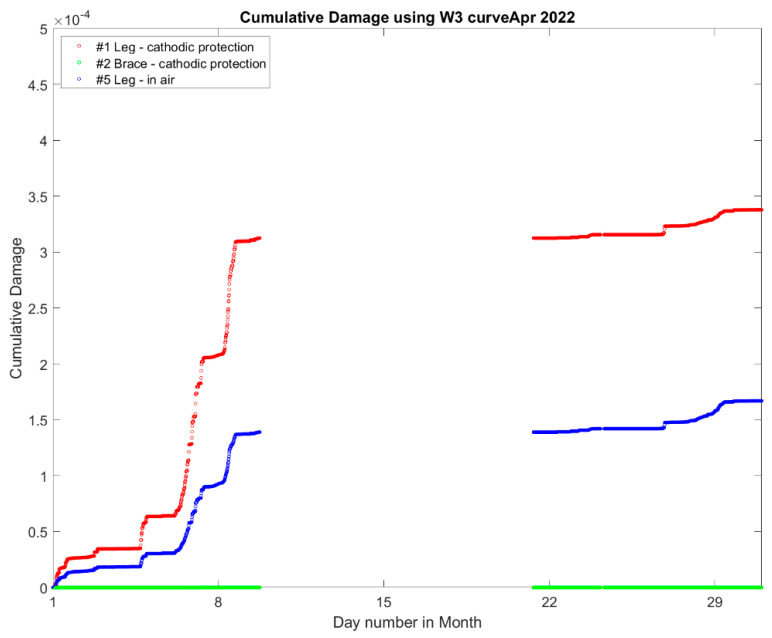
Damage accumulation at the jacket hotspots in April 2022.

**Figure 22 sensors-24-03009-f022:**
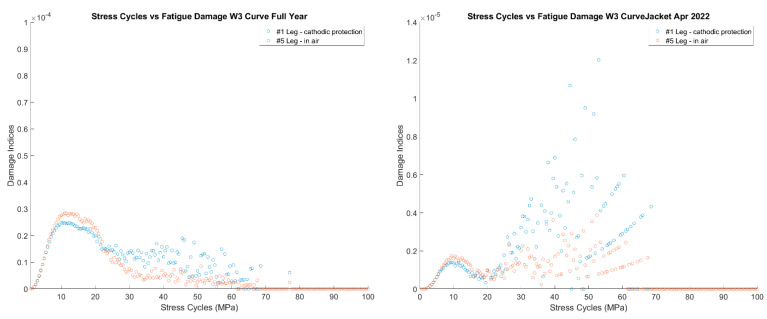
Damage by stress-cycle range for 1-year monitoring and April 2022.

**Figure 23 sensors-24-03009-f023:**
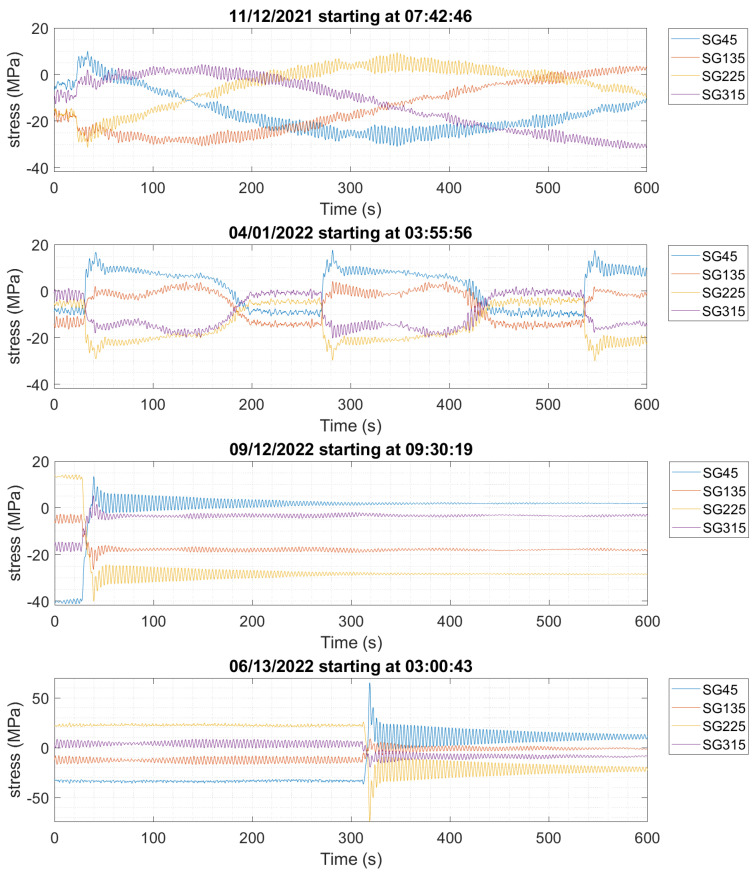
Stress time histories during 10-min data windows impose relatively high damage to the OWT on 12 November 2021 starting at 7:42 a.m., 1 April 2022 starting at 3:55 a.m., 12 September 2022 starting at 9:30 a.m., and 13 June 2022 starting at 3:00 a.m.

**Figure 24 sensors-24-03009-f024:**
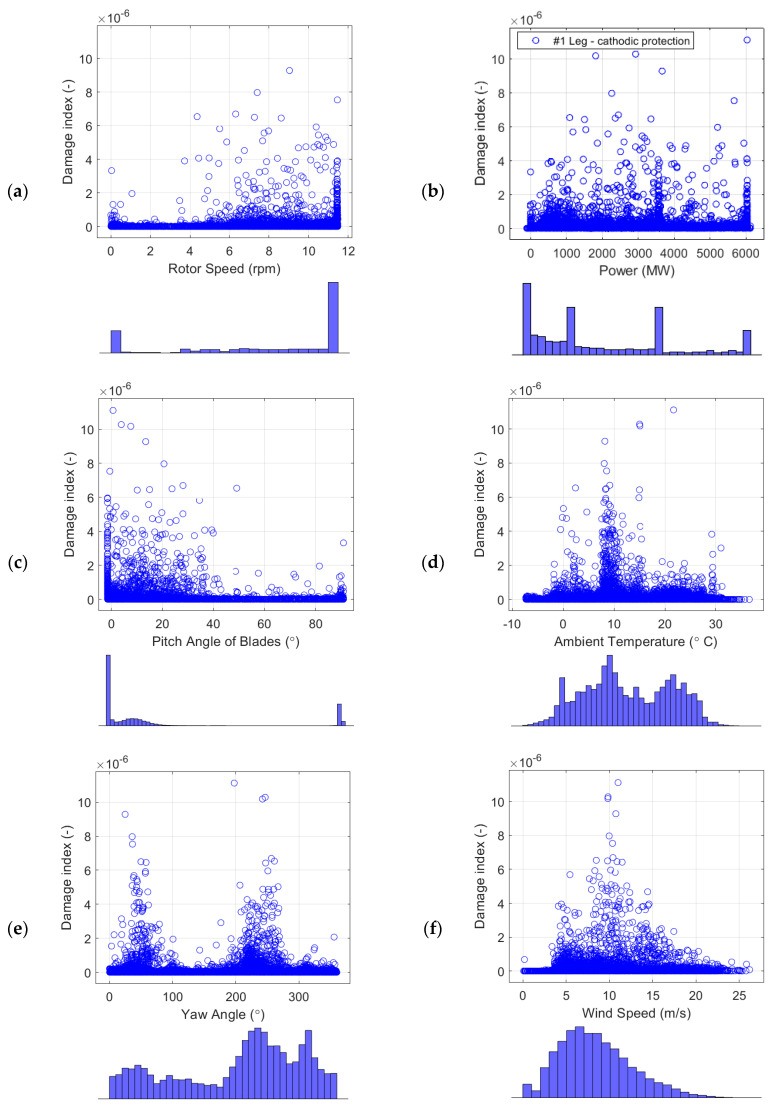
Effects of (**a**) rotor speed, (**b**) power yaw angle, (**c**) pitch angle of each blade, (**d**) ambient temperature, (**e**) yaw angle, and (**f**) wind speed on the damage indices of the jacket leg #1 with cathodic protection in seawater.

**Figure 25 sensors-24-03009-f025:**
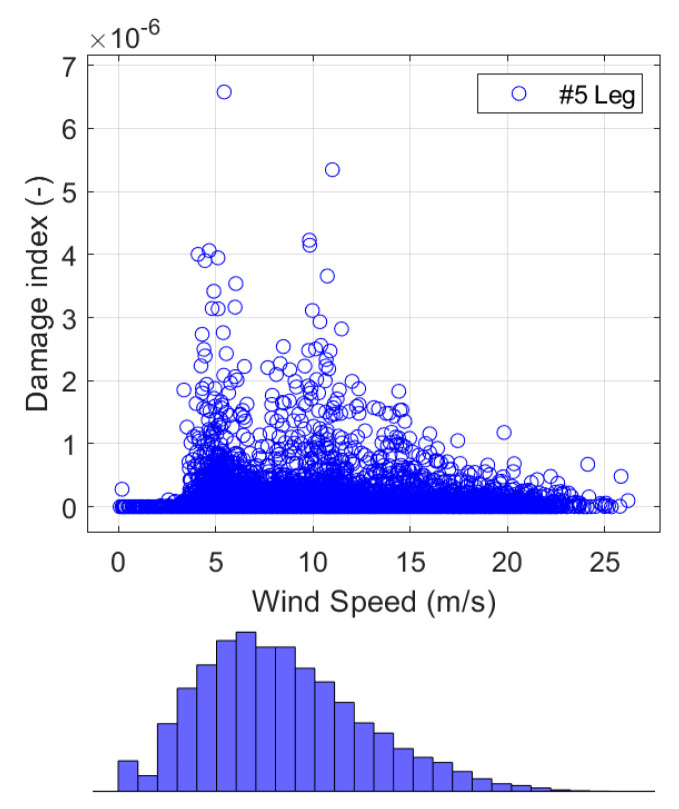
Effects of wind speed on the damage indices of jacket leg #5 in the splash zone.

**Figure 26 sensors-24-03009-f026:**
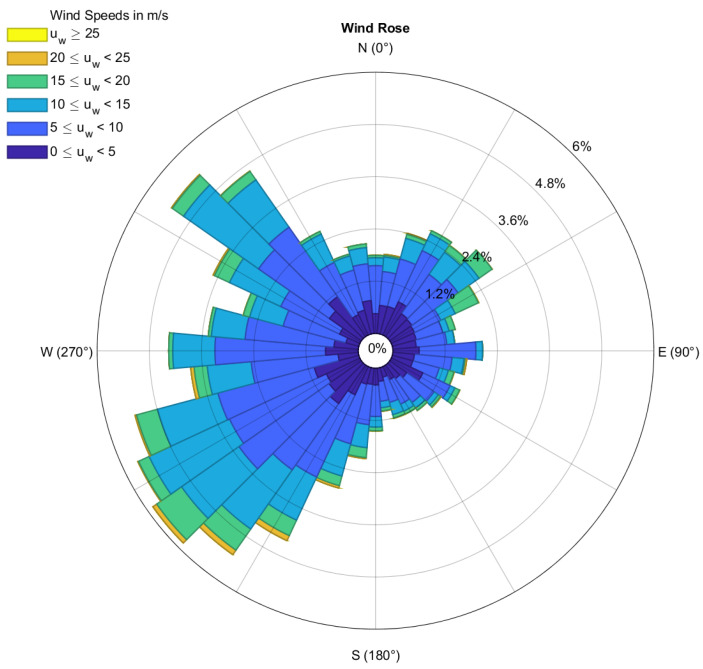
Wind rose for the block island turbine from 1 November 2021 to 1 October 2022. (The MATLAB function for plotting a wind rose can be found in [52]).

**Figure 27 sensors-24-03009-f027:**
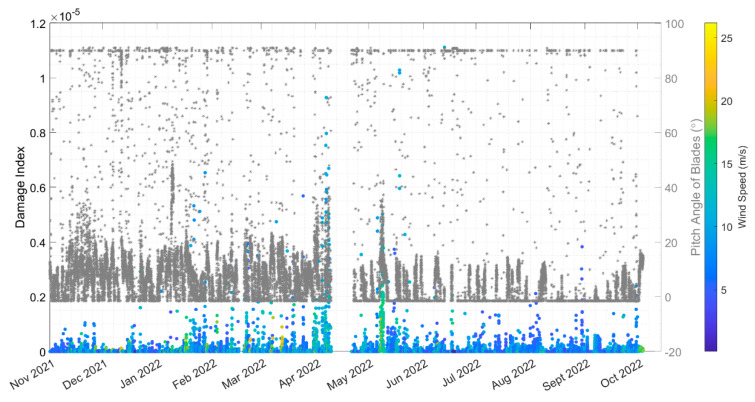
Damage index for the jacket joint #1 vs. pitch angle of the blades during 1 year of monitoring from November 2021 to October 2022.

**Figure 28 sensors-24-03009-f028:**
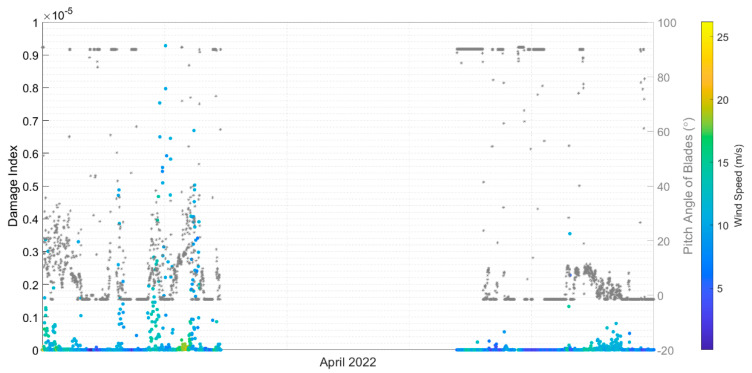
Damage index for the jacket joint #1 vs. the pitch angle of blades during April 2022.

**Figure 29 sensors-24-03009-f029:**
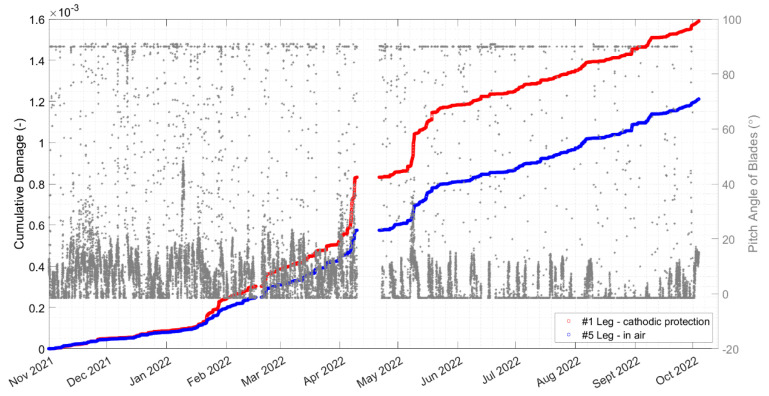
Cumulative damage of the jacket joint #5 vs. the pitch angle of blades during 1 year of monitoring from 1 November 2021 to 1 October 2022.

**Table 1 sensors-24-03009-t001:** The number of missing SG and acceleration time history 10-min datasets used each month.

Month	November 2021	December 2021	January 2022	February 2022	March 2022	April 2022	May 2022	June 2022	July 2022	August 2022	September 2022	October 2022	Total
**# Datasets**	11	18	7	416	116	1690	8	11	1	130	5	1028	3441

**Table 2 sensors-24-03009-t002:** Stresses at several sections in the legs and braces of the BIWF jacket (MPa/1 MN).

	Joint	1	2	3	4	5
Angle						
**0°**		−55.5	±9.5	45.1	±8.6	52.4
**15°**		−65.0	±16.8	53	±9.8	57.0
**30°**		−66.9	±15.3	59.6	±11.4	83.9
**45°**		−70.3	±12.9	61.7	±11.7	84.1

**Table 3 sensors-24-03009-t003:** Jacket node stresses obtained from the FE model due to 1 MN thrust force at the RNA level.

Joint	Angle (°)	FA Mode S11 Max (MPa/1MN)	In Splash Zone/In Seawater	Cathodic Protection
1	45	−70.3	Seawater	Yes
2	45	16.8	Seawater	Yes
3	45	61.7	Seawater	Yes
4	15	11.7	Seawater	Yes
5	45	84.1	Splash zone	No

**Table 4 sensors-24-03009-t004:** S-N curves for various joint weld types and the environment, adapted from DNVGL-RP-C203.

S-N Curve	Environment	$\log{\bar{a}}_{1}$	$b_{1}$	$\log{\bar{a}}_{2}$	$b_{2}$
B1	In Air	N ≤ 10^7^ cycles		N > 10^7^ cycles	
		15.117	4	17.146	5
C1	In Air	12.499	3	16.081	5
Tubular	In Air	12.48	3	16.13	5
W3	In Air	10.97	3	13.617	5
Tubular	Cathodic Protection	N ≤ 1.8∙10^6^ cycles		N > 1.8∙10^6^ cycles	
		12.18	3	16.13	5
W3	Cathodic Protection	N ≤ 10^6^ cycles		N > 10^6^ cycles	
		10.57	3	13.617	5

**Table 5 sensors-24-03009-t005:** Ratios of jacket hotspots to the tower strain mode shapes for the first u or v mode.

Ratios	#1 Leg	#2 Brace	#3 Leg	#4 Brace	#5 Leg
** *r_u_* **	−0.67	0.08	0.66	−0.15	0.65
** *r_v_* **	−0.66	−0.04	0.66	−0.04	0.66

**Table 6 sensors-24-03009-t006:** Estimated lifespan tower in years.

	Estimated Lifetime Adjusted			
**Curve**	SG45	SG135	SG225	SG315
**B1 Air**	Below FL	Below FL	Below FL	Below FL
**C1 Air**	7.75 × 10^4^	8.55 × 10^4^	8.85 × 10^4^	6.71 × 10^4^

**Table 7 sensors-24-03009-t007:** Damage indices for the jacket joints using the modal expansion method during 1 year of monitoring from 1 November 2021 to 30 October 2022 (no DFF used).

Damage During One Year of Monitoring			
Joint	1	2	5
**Tubular joint with cathodic protection**	7.2 × 10^−6^	4.7 × 10^−9^	-
**Tubular joint in air**	-	-	7.4 × 10^−6^
**W3 with cathodic protection**	0.0017	1.5 × 10^−6^	-
**W3 in air**	-	-	0.0013

**Table 8 sensors-24-03009-t008:** Service-life comparison between the extrapolation of 1 y monitoring and design values using DFF. DFF = 2 and 3 are used for the environment of seawater and splash zone, respectively.

Lifetime Estimation (Years)					
Joint	Environment	DFF	1-Year Monitoring (Static Calc)	1-Year Monitoring (Modal Expansion)	Design
1	W3 with cathodic protection	3	65	196	26
2	W3 with cathodic protection	3	35,000 *	222,000 *	76
5	W3 in air	2	62	386	50

* Extremely low fatigue demand.

**Table 9 sensors-24-03009-t009:** SCADA for the highest damage index during 1-year monitoring caused by the dataset 2022_06_13_000043 (13-Jun-2022 at 3:00:43 am), one SCADA before and after that dataset. Note that the offset is due to the clock lag in the DAQ system (for about 10 mins).

SCADA TimeStamp	Wind Speed (m/s)	Power (kW)	Yaw Angle (°)	Pitch Angle (°)	Damage Index (-)
**13 June 2022 03:00:00**	11.04	6034	197.51	1.23	6.9 × 10^−9^
**13 June 2022 03:10:00**	11.00	6030	197.51	0.66	1.2 × 10^−5^
**13 June 2022 03:20:00**	10.40	131	197.51	86.99	0

## Data Availability

The datasets presented in this article are under the NDA and, thus, are not available for sharing.

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
