# Peer review of "Fatigue Analysis of a Jacket-Supported Offshore Wind Turbine at Block Island Wind Farm"

_sensors, 2024, doi:10.3390/s24103009_

Round 1

Reviewer 1 Report

Comments and Suggestions for Authors

The work is a study on the fatigue analysis of a jacket-supported offshore wind turbine at the Block Island Wind Farm. It assesses the remaining fatigue life of the support structure for a GE Haliade 6-MW fixed-bottom jacket offshore wind turbine. The study uses stress time histories at instrumented and non-instrumented locations, and two validated finite element models are utilized for assessing the stress cycles. The modal expansion method and a simplified approach using static calculations of the responses are employed to estimate the stress at the non-instrumented locations, known as virtual sensors. I have the following comments and questions regarding this paper:

1. Clarify the methodology used for assessing stress cycles and fatigue damage, including the specific parameters and criteria used for the modal expansion method and the simplified static approach.

2. Besides conventional analysis methods, deep learning approaches have garnered considerable attention in structural response analysis. These emerging trends should be reflected in the introduction section of the paper. For instance: Engineering Structures, 2023, 297: 117027. Computational Mechanics, 2021, 67: 207-230.

3. Provide more detailed information on the hysteresis method used for evaluating tower fatigue and the rainflow counting method for evaluating stress ranges and cycles.

4. Include a discussion on the potential implications of the findings for the design and maintenance of offshore wind turbines, and how the results could contribute to improving structural health monitoring and damage prognosis in the industry.

5. Consider adding a section on the limitations of the study, including any assumptions made in the analysis and potential areas for future research to address these limitations.

Comments on the Quality of English Language

Minor editing of English language required.

Author Response

The authors would like to thank the Reviewers for their constructive feedback. The manuscript has been revised accordingly based on the comments of the Reviewers. A point-by-point response to the Reviewers’ comments is provided below.

  1. Clarify the methodology used for assessing stress cycles and fatigue damage, including the specific parameters and criteria used for the modal expansion method and the simplified static approach.

Answer:

The methodology to assess fatigue damage is rainflow counting, which is provided by the ASTM E1049-85 standard [1]. The rainflow counting is available in a Matlab function named 'rainflow', which is used in this paper.

In the application of modal expansion, we use a FE model to extract the displacement mode shape, then using equation 19, strain mode shapes are calculated from the displacement mode shapes. The stress response at any hotspot is assessed using the strain mode shapes and the strain measurements at the tower base. The parameters related to filtering the strain measurements have been discussed.

The following text is added to Section 4.

"The first step in assessing fatigue damage is to count the stress cycles and determine the stress ranges from a window of stress response data. According to the ASTM E1049-85 standard [39], several counting methods are related to fatigue damage assessment, and rainflow counting is used in this paper."

  1. Besides conventional analysis methods, deep learning approaches have garnered considerable attention in structural response analysis. These emerging trends should be reflected in the introduction section of the paper. For instance: Engineering Structures, 2023, 297: 117027. Computational Mechanics, 2021, 67: 207-230.

Answer:

This is true and thank you for pointing this out. The following has been added to the Introduction.

Besides conventional structural analysis methods, neural networks can predict structural responses nowadays. The emerging trends can be found in [2, 3].

  1. Provide more detailed information on the hysteresis method used for evaluating tower fatigue and the rainflow counting method for evaluating stress ranges and cycles.

Answer:

A brief discussion on the hysteresis method that has been added to Section 4 is provided here. 

Hysteresis filtering was used based on recommendations from online resources related to fatigue analysis found in both Siemens and MATLAB documentation [4]. Hysteresis filtering works by removing reversals below a minimum threshold from the time series. For this analysis, the minimum threshold is set at 0.5 MPa. After utilizing hysteresis filtering to remove these very small cycles, peak-valley filtering is used to identify the local minima and maxima in the time series. Hysteresis filtering and peak-valley filtering are found in the 'findTurningPts' function from the MATLAB resource on fatigue analysis. Hysteresis and peak-valley filtering reduce the low-stress cycles to 0.5 MPa, and the peak and valleys of the stress time history become clearer. However, hysteresis filtering is insufficient as it does not have a frequency component to reduce noise, and the low-stress cycles (e.g., 1 MPa) remain. So, an FIR filter added to the raw measurement is needed to remove low-frequency noises. In this study, both FIR and hysteresis filtering are used to assess the fatigue damage.

Figure 19 b shows how the original data seen in blue is then simplified to only include the local peaks and valleys greater than 0.5 MPa. Rainflow counting could be used on the unfiltered dataset but doing so would result in counting millions of low-stress cycles (e.g., 0.1 MPa) that do not have any meaningful impact on fatigue.

The following text has been added to Section 5.4.1.

"After hysteresis and peak-valley filtering the original data, as shown in solid blue line with spikes and multiple stress values at peaks and valleys, is simplified to only include the single local peaks and valleys greater than 0.5 MPa. Rainflow counting could be used on the unfiltered dataset but doing so would result in counting millions of low-stress cycles (e.g., 0.1 MPa) that do not have any meaningful impact on fatigue."

  1. Include a discussion on the potential implications of the findings for the design and maintenance of offshore wind turbines, and how the results could contribute to improving structural health monitoring and damage prognosis in the industry.

Answer:

The following text has been added to the Conclusions section:

"As damage to the jacket leg is found to be larger than that in the splash zone and maintenance is inaccessible underwater, particular fatigue analysis and capacity should be considered when designing the foundation elements underwater. The hotspots at the splash zone are assumed to be coated for the remaining lifetime of the turbine, and they should be maintained coated; otherwise, the damage would be greater than the estimated value in this paper. Moreover, to have fewer high-stress cycle ranges in the foundation hotspots, the number of shutdowns and startups should be minimized during the maintenance of offshore wind turbines."

  1. Consider adding a section on the limitations of the study, including any assumptions made in the analysis and potential areas for future research to address these limitations.

Answer:

The following has been added to Section 6.

"6.1. Limitations and future work

In assessing the fatigue damage, the effect of the thickness of a plate and the stress concentration factor (SCF) are not considered when using the S-N curves. The mean stress value is also dismissed from the fatigue damage calculation. According to DNVGL-RP-C203, if part of the stress cycle is in compression, the stress ranges may be reduced by up to 20% before entering the S-N curve. So, considering the mean stress, thickness effect, and SCF is recommended for future work.

Furthermore, this work has linearly extrapolated the 1-year fatigue damage to the 25-year lifetime of the OWT. For having an accurate assessment of the fatigue damage, a sophisticated model, such as a neural network, can be trained in future work to predict the fatigue damage using the forecast from climate models for remaining design life for the BIWF site. "

References:

  1. ASTM-E1049-85, Standard Practices for Cycle Counting in Fatigue Analysis. 2017.
  2. Xu, Z., et al., Physics guided wavelet convolutional neural network for wind-induced vibration modeling with application to structural dynamic reliability analysis. Engineering Structures, 2023. 297: p. 117027.
  3. Liu, Y., et al., HiDeNN-FEM: a seamless machine learning approach to nonlinear finite element analysis. Computational Mechanics, 2023. 72(1): p. 173-194.
  4. MathWorks. Practical Introduction to Fatigue Analysis Using Rainflow Counting. Available from: https://www.mathworks.com/help/signal/ug/practical-introduction-to-fatigue-analysis-using-rainflow-counting.html#mw_rtc_PracticalIntroToFatigueAnalysisUsingRainflowCountingExample_M_68F0E3B8.

Reviewer 2 Report

Comments and Suggestions for Authors

A review of

Fatigue Analysis of a Jacket-Supported Offshore Wind Turbine at Block Island Wind Farm

by Nasim Partovi-Mehr et al.

This paper reported a field test of an Offshore wind turbine (OWT) in the Block Island Wind Farm. The remaining fatigue life of the support structure for a GE Haliade 6-MW fixed-bottom jacket offshore wind turbine is estimated using the virtual sensing and modal expansion method. In general, this paper is well-written. The following are my concerns and suggestions which may help the authors in further improving the readability of the paper:

Although this paper has presented the method used for the correction of strain gauge zeros, another issue related to strain gauges is that they are not suitable for long-term sensing. A drift of signal will take place after a certain time due to the aging of the connection between the strain gauges and the host structure. How does this paper address this issue?

According to Figure 3, the FE model established in SAP2000 is a simplified model without consideration of the blades and the machines at the top of the OWT. Will this simplification influence the accuracy of the modal expansion method?

Pay more attention to the format of the paper. I think the word “where” after each equation shall be at the very front of each line. The space before “where” shall be removed.

Author Response

The authors would like to thank the Reviewers for their constructive feedback. The manuscript has been revised accordingly based on the comments of the Reviewers. A point-by-point response to the Reviewers’ comments is provided below.

  • Although this paper has presented the method used for the correction of strain gauge zeros, another issue related to strain gauges is that they are not suitable for long-term sensing. A drift of signal will take place after a certain time due to the aging of the connection between the strain gauges and the host structure. How does this paper address this issue?

Answer:

This is a good point raised by the reviewer. The strain gauges (SGs) undergo the correction process every month. This process is discussed in 2.2 resulting in correcting the strain measurements 12 times (once a month) during the 1-year monitoring. We fitted 12 circles, similar to the circles in Figure 7. The radius for each month's circle is checked to stay constant, to represent the self-weight moment of the RNA and is not observed to change over time. We strive to keep the center of the circles at zero. Therefore, by checking the circles’ zero centers and radii, we can adjust the strain measurements to account for drift over time.

The following text has been added to Section 5.1.

“It is important to note that SGs may not be suitable for long-term virtual sensing, as a drift in strain measurements can occur. The SG measurements undergo a correction process every month to address this issue in this study. In addition to the circle in Figure 7, eleven more circles are fitted for different months during the 1-year monitoring data. The radius for each month's circle is checked to stay constant, as it represents the self-weight moment of the RNA and is not changed significantly over time. The center of the circles is kept at zero. Therefore, strain measurements are adjusted by checking the circles’ zero centers and radii to account for possible drift over time.”  

  • According to Figure 3, the FE model established in SAP2000 is a simplified model without consideration of the blades and the machines at the top of the OWT. Will this simplification influence the accuracy of the modal expansion method?

Answer:

This is a valid inquiry about accuracy of the model. The FE model in SAP2000 does not have the elements of the blades and machinery at the top of the support structure, but the moment of inertia and their mass are used in the model. Discretized elements for the blades and RNA will have negligible effects on the first bending mode for the tower and support structure. Therefore, their addition will not affect the modal expansion results at any point on the tower or jacket. However, if virtual sensing was needed at any location on blades or RNA, their detailed modal response (mode shapes) would be needed and simplification of them in the model would adversely affect the results.

  • Pay more attention to the format of the paper. I think the word “where” after each equation shall be at the very front of each line. The space before “where” shall be removed.

Answer:

Thank you for bringing this to our attention. The spaces have been removed in the revised version of the paper.

Reviewer 3 Report

Comments and Suggestions for Authors

Dear authors

Thank you for the interesting paper. However, I would argue that the paper is too long. The paper is 43 pages long and I do not think the content of the paper justifies this many pages. In the paper, there are a lot of redundant information or information that should be known to the reader and should have been a reference instead.

-      Line 252 gives the value of EI/y while it states the induvial values in the sentence before. That is not needed. If the reader wants to know EI/y then the reader can calculate it themselves.

-      For instance, equation 28 is not needed.

-      Likewise, the entire part of calculating strain from a beam element, page 13-17, is not needed and it could have been a few lines of text instead.

The scientific message of the paper is also a bit unclear and the paper misses a clear common thread. It seems to be an application study of existing techniques on a wind turbine with one year of data. Then it also seems that the authors want to show too much of their analyses. Section 5 is too long and confusing. It is 23 pages long with 31 figures and 5 tables. The section holds a lot of results, which seem to be unrelated to each other. It needs to be significant shortened, and the scope should be reduced/clearer, so your message of the section is clear to the reader. I would recommend removing the subsection on the numerical study and focus on the experimental data instead.

I would recommend shortening and tightening the paper, so the message is clearer.

Other comments:

The term “offshore wind” is used in the paper; however, this term has multiple definitions. I would personally use the term for the wind itself and not related to wind turbines. So, I would advise to use the term “offshore wind turbine” or to define the term.

Check your reference list. Some of the references are incorrectly listed where surnames are swapped with first names. This reviewer saw this for ref. 4 & 31. Please ensure that all references have the same setup.

The term “virtual sensing” is not defined. Please insert explanation of this.

There is a paragraph about Ramboll in the introduction without any references. Please insert some for this part. The reference could be:

-      Tygesen, UT, Jepsen, MS, Vestermark, J, Dollerup, N, & Pedersen, A. "The True Digital Twin Concept for Fatigue Re-Assessment of Marine Structures." Proceedings of the ASME 2018 37th International Conference on Ocean, Offshore and Arctic Engineering

Consider the use of acronym and abbreviation. There is a lot of them in this paper, and they are not all used very much. But the reader will have to remember all of them when reading the text. Acronyms should ease reading but here the extensive use makes the text more difficult to read since the reader must remember a lot of acronyms. So, I would advise to remove some of the acronyms that is not used frequently in the text.

What is “m” in equation 1 and 2? “m” is used as Wöhlers constant in the introduction so it would seem that you take the power m of the strain measurements.

There should be no indent after an equation when you add information to the equation. This could be “where”.

Lines 467-478 suddenly defines variables in a new way. This reviewer prefers a consistent style for defining variables.

Line 654 holds an incorrect reference to a section.

Author Response

The authors would like to thank the Reviewers for their constructive feedback. The manuscript has been revised accordingly based on the comments of the Reviewers. A point-by-point response to the Reviewers’ comments is provided below.

  • Thank you for the interesting paper. However, I would argue that the paper is too long. The paper is 43 pages long and I do not think the content of the paper justifies this many pages. In the paper, there are a lot of redundant information or information that should be known to the reader and should have been a reference instead.

Answer:

Thank you for suggesting the sections where the paper can be trimmed. The revised paper has been shortened by addressing the reviewer’s suggestions.

  • Line 252 gives the value of EI/y while it states the induvial values in the sentence before. That is not needed. If the reader wants to know EI/y then the reader can calculate it themselves.

Answer:

The EI/y has been removed in the revised paper.

  • For instance, equation 28 is not needed.

Answer:

While Equation 28 is not independently needed as it is a combination of two past Equations (25 and 27), we believe having this equation helps with readability and the explanation that comes after the equation.

  • Likewise, the entire part of calculating strain from a beam element, page 13-17, is not needed and it could have been a few lines of text instead.

Answer:

The section on computation of strains in beam elements (pages 13-14 of the original manuscript) is trimmed in the revised manuscript.

  • The scientific message of the paper is also a bit unclear and the paper misses a clear common thread. It seems to be an application study of existing techniques on a wind turbine with one year of data. Then it also seems that the authors want to show too much of their analyses. Section 5 is too long and confusing. It is 23 pages long with 31 figures and 5 tables. The section holds a lot of results, which seem to be unrelated to each other. It needs to be significant shortened, and the scope should be reduced/clearer, so your message of the section is clear to the reader. I would recommend removing the subsection on the numerical study and focus on the experimental data instead. I would recommend shortening and tightening the paper, so the message is clearer.

Answer:

Thank you for your feedback. The numerical results section has been removed from the paper. The paper has been shortened in the Results Section. For example, Table 6, and seven figures have been removed or merged with other figures. We believe this revision helps with a better and more concise presentation of findings to the reader.

Other comments:

  • The term “offshore wind” is used in the paper; however, this term has multiple definitions. I would personally use the term for the wind itself and not related to wind turbines. So, I would advise to use the term “offshore wind turbine” or to define the term.

Answer:

The term offshore wind was replaced by either OWT or offshore wind industry in the revised paper.

  • Check your reference list. Some of the references are incorrectly listed where surnames are swapped with first names. This reviewer saw this for ref. 4 & 31. Please ensure that all references have the same setup.

Answer:

The references have been checked and the errors in authors names are corrected in this revision.

  • The term “virtual sensing” is not defined. Please insert explanation of this.

Answer:

The following text has been added to Section 1.

“Virtual sensing is the process of estimating the response of a system at locations that are difficult to measure through methods that use a combination of models and existing physical sensor data.”

  • There is a paragraph about Ramboll in the introduction without any references. Please insert some for this part. The reference could be:

-      Tygesen, UT, Jepsen, MS, Vestermark, J, Dollerup, N, & Pedersen, A. "The True Digital Twin Concept for Fatigue Re-Assessment of Marine Structures." Proceedings of the ASME 2018 37th International Conference on Ocean, Offshore and Arctic Engineering

Answer:

The suggested reference has been added to the introduction as reference 17.

  • Consider the use of acronym and abbreviation. There is a lot of them in this paper, and they are not all used very much. But the reader will have to remember all of them when reading the text. Acronyms should ease reading but here the extensive use makes the text more difficult to read since the reader must remember a lot of acronyms. So, I would advise to remove some of the acronyms that is not used frequently in the text.

Answer:

Acronyms LCoE, O&M, RBI, BSEE, NGI, GE, and FLS have been removed and their full names used in the text.

  • What is “m” in equation 1 and 2? “m” is used as Wöhlers constant in the introduction so it would seem that you take the power m of the strain measurements.

Answer:

The following text has been added to the revised paper.

“Superscript m in Equations 1-2 stands for the measurement.”

Other “m” notations in the paper have been replaced with other variables to avoid confusion. For example, the “m” term defining negative inverse slope of the S-N curve has been relabeled as “b” in Equations (33) and (34). Labels b1 and b2 have been used instead of m1 and m2 in Table 4. Mass of the RNA in Equation (16) has been defined by mRNA.

  • There should be no indent after an equation when you add information to the equation. This could be “where”.

Answer:

The indentations before “where” after equations have been removed.

  • Lines 467-478 suddenly defines variables in a new way. This reviewer prefers a consistent style for defining variables.

Answer:

The variables' definitions have been changed in the revised paper to be consistent with earlier definitions.

  • Line 654 holds an incorrect reference to a section.

Answer:

The section reference is edited in the revised paper as follows.

“The damage indices are discussed further in Section 5.3.2.”

Round 2

Reviewer 1 Report

Comments and Suggestions for Authors

The authors have addressed most of the concerns of the reviewer. The manuscript in its current form can be recommended for publication.